# Simulations of black carbon (BC) aerosol impact over Hindu-Kush Himalayan sites: validation, sources, and implications on glacier runoff

Sauvik Santra[1], Shubha Verma[1], Koji Fujita[2], Indrajit Chakraborty[1], Olivier Boucher[3], Toshihiko Takemura[4], John F. Burkhart[5], Felix Matt[5], and Mukesh Sharma[6]

[1]Department of Civil Engineering, Indian Institute of Technology Kharagpur, India
[2]Graduate School of Environmental Studies, Nagoya University, Nagoya, Japan
[3]Institut Pierre-Simon Laplace, Centre National de la Recherche Scientifique / Sorbonne Université, 75252 Paris Cedex 05, France
[4]Research Institute for Applied Mechanics, Kyushu University, Fukuoka, Japan
[5]Department of Geosciences, University of Oslo, Norway
[6]Department of Civil Engineering, Indian Institute of Technology Kanpur, India

**Correspondence:** Shubha Verma (shubha@iitkgp.ac.in)

**Abstract.** We estimated the black carbon (BC) concentration over the Hindu Kush Himalayan region (HKH), its impact on snow-albedo reduction and sensitivity on annual glacier runoff over the identified glaciers. These estimates were based on free-running aerosol simulations ($freesimu$) and constrained aerosol simulations ($constrsimu$) from an atmospheric general circulation model, combined with numerical simulations of glacial mass balance model. BC concentration estimated from
5 $freesimu$ performed better over higher altitude (HA) HKH stations than that over lower altitude (LA) stations. The estimates from $constrsimu$ mirrored well the measurements when implemented for LA stations. Estimates of the spatial distribution of BC concentration in the snowpack ($BC_c$) over the HKH region led to identifying a hot-spot zone located around Manora peak. Among glaciers over this zone, $BC_c$ (>60 $\mu$g kg$^{-1}$) and BC-induced snow-albedo reduction ($\approx$5%) were estimated explicitly being high during the pre-monsoon for Pindari, Poting, Chorabari, and Gangotri glaciers (which are major sources of fresh water for the Indian sub-continent). The rate of increase of $BC_c$ in recent years (i.e. over the period 1961–2010) was,
however, estimated being the highest for the Zemmu glacier. Sensitivity analysis with glacial mass balance model indicated the increase in annual runoff from debris-free glacier area due to BC-induced snow albedo reduction (SAR) corresponding to $BC_c$ estimated for the HKH glaciers was 4% – 18%, with the highest being for the Milam and Pindari glacier. The rate of increase in annual glacier runoff per unit BC-induced percentage SAR was specifically high for Milam, Pindari, and Shunkalpa glacier.
The source-specific contribution to atmospheric BC aerosols by emission sources led to identifying the potential emission source being primarily from the biofuel combustion in the Indo-Gangetic plain south to 30°N, but also from open burning in a more remote region north to 30°N.

# 1 Introduction

Aerosols are particles in the atmosphere known to impact Earth's climate directly as well as indirectly (Foster, 2007). Aerosols tend to scatter or absorb insolation and thus affect the radiation budget likewise Ramanathan et al. (2001a, b). Among aerosol constituents, black carbon (BC) is a major absorbing aerosol constituent produced from incomplete combustion of biomass, biofuel, and fossil fuel (Bond et al., 2013). Black carbon impacts the climate through a direct effect by absorbing sunlight (Ramanathan and Carmichael, 2008) and an indirect effect through cloud alterations in precipitation efficiency (Lohmann and Feichter, 2005). Another impact due to absorbing aerosols such as BC exists on the cryosphere by altering the ablation rate of ice and snow (Flanner et al., 2007; Painter et al., 2007). Deposited black carbon over snow enhances absorption of solar radiation, darkens the upper mixing layers of the snowpack thereby reducing the snow albedo (Warren and Wiscombe, 1980) and leading to accelerated melting of snow (Jacobson, 2004; Flanner et al., 2007). The Himalayan region has been exposed to particulate pollution and deposition of BC as reported in observational studies at high altitude Himalayan stations (Bonasoni et al., 2012, 2010a, b; Marinoni et al., 2010; Nair et al., 2013; Jacobi et al., 2014). Based on the atmospheric BC measurements at stations (e.g. NCOP, Hanle) over the Hindukush-Himalayan (HKH) region, radiative forcing due to BC and the BC-induced snow albedo reduction (SAR) for estimated BC concentration in snow over the stations was reported in recent studies (Nair et al., 2013; Yasunari et al., 2010). Additionally, the premature ablation of snowpack over the Himalayan region (at Khumbu valley, Nepal), which brought forward the melting of snowpack by 17 to 27 days has also been inferred due to BC-induced snow-albedo reduction (SAR) based on snowpack modelling (Jacobi et al., 2014). Further, the impact of BC aerosols on the snowpack, surface radiation, and temperature changes over the HKH region based on simulations in global climate models suggest a considerable impact of anthropogenic forcing over the HKH region as compared to that over the Tibetan Plateau. However, the ability of coarse-gridded models to simulate adequately the snow depth and thereby the BC concentration in snow and atmospheric BC radiative forcing is limited (Menon et al., 2010; Ménégoz et al., 2014; Qian et al., 2015).

The influence of the albedo change on the glacial mass balance due to an excess and preponed snow melting and thereby the glacier runoff is expected to impact the downstream hydrology (Fujita, 2007; Matt et al., 2018; Sakai and Fujita, 2017). This impact is specifically of concern for the HKH region as the Himalayan glaciers are the source of major rivers in South Asia namely Ganges, Indus, Yamuna, and Brahmaputra (also known as Tsangpo). The inaccessible terrain and severe weather conditions in the higher Himalayan region hinder the measurement of atmospheric BC concentration and BC concentration in snow (Ming et al., 2009) at regular spatial as well as temporal intervals. The measured data may thus serve as a location and time specific primary data and not a representative sample of the regional distribution. The simulated BC concentration, using atmospheric chemical transport models (CTMs), which is validated by measurements, can be utilized to predict the spatial mapping of BC distribution over the HKH region. In order to spatially map as adequately as possible the estimates of atmospheric BC concentration and BC concentration in snow including the corresponding SAR over the HKH region, an integrated approach merging the relevant information from observations with a relatively consistent atmospheric chemical transport model estimates is applied in the present study.

In the present study, we evaluate BC concentration estimated from the free running ($freesimu$) aerosol simulations us-

ing Laboratoire de Météorologie Dynamique atmospheric General Circulation Model (LMDZT-GCM) and Spectral Radiation Transport Model for Aerosol Species (SPRINTARS) over the HKH region. This evaluation includes a comparison of the simulated BC concentration with observations (refer to Sections 3.1 and 3.2), thereby leading to identifying the most consistent $freesimu$ estimates with observations out of the three $freesimu$. The comparison is done with the available observations at HKH sites (Nair et al., 2013; Marinoni et al., 2010; Babu et al., 2011) for winter (monthly average of December, January, and February) and pre-monsoon (monthly average of March, April, and May) season, at locations, classified as low-altitude (LA) stations (e.g. Nainital, Kullu, and Dehradun, refer to Figure:1(a)), which are in close proximity to emission sources; and high-altitude (HA) stations (e.g. Hanle, NCO-P, and Satopanth, Figure:1(b)), which are relatively remotely located and mostly influenced by transport of aerosols. Constrained aerosol simulation ($constrsimu$) is also formulated (discussed in Section 2.2) using the simulated aerosol characteristics from the identified $freesimu$, and $constrsimu$ estimates are evaluated over the LA stations. Details of the simulation used can be found in Table 2. Impact of BC concentration in the snow ($BC_c$) on SAR is estimated (Section 2.3); and numerical simulations of annual glacier runoff height and snow-albedo are carried out using glacial mass balance model (Section 2.4) to evaluate the impact of BC-induced SAR on increase in annual snowmelt runoff from glaciers.

**Table 1.** Description of glaciers considered in this study

| Glacier Name (abbreviation) | Lat. ($°N$) | Lon. ($°E$) | Area ($km^2$) | Elevation (m a.s.l.) | RDCA* (%) | Recession Rate (m y$^{-1}$) | Reference |
|---|---|---|---|---|---|---|---|
| Bara Shigri (BS) | 32.166 | 77.695 | 112.4 | 3931-6438 | 17.8 | 30.0 | Sangewar and Kulkarni (2011) |
| Chorabari (CB) | 30.786 | 79.046 | 4.6 | 3926-6717 | 28.1 | 6.8 | Mehta et al. (2014) |
| Gangotri (GG) | 30.800 | 79.149 | 136.9 | 3995-7062 | 25.9 | 34.0 | Naithani et al. (2001) |
| Milam (ML) | 30.520 | 80.050 | 59.2 | 3552-6989 | 32.5 | 25.5 | Raj (2011) |
| Pindari (PD) | 30.301 | 80.007 | 11.4 | 3669-6568 | 7.4 | 23.4 | Vohra (1981) |
| Poting (PT) | 30.235 | 80.133 | 4.5 | 3665-5639 | 15.0 | 5.1 | Vohra (1981) |
| Shunkalpa (SK) | 30.378 | 80.320 | 34.3 | 3952-6527 | 24.1 | 6.8 | Vohra (1981) |
| Sonapani (SP) | 32.421 | 77.355 | 6.7 | 3805-5562 | 0.0 | 15.9 | Vohra (1981) |
| Zemmu (ZM) | 27.743 | 88.238 | 68.6 | 4178-7402 | 35.6 | 19.8 | Vohra et al. (1993) |

[*]RDCA = Ratio of debris-covered area

In order to examine the source of BC aerosols over the HKH region due to emissions from combustion sectors and that from nearby or remote region, source- and region-tagged BC simulation in LMDZT-GCM (Verma et al., 2011) is also evaluated.

Hence, the specific objectives of this study includes evaluation of (1) atmospheric BC concentration estimated from $freesimu$ and $constrsimu$ over the HKH region, (2) spatial distribution of $BC_c$ and identification of hot-spot zone and glaciers located in this zone, (3) impact of $BC_c$ on SAR over the glaciers and its potential to increase the annual glacier runoff through numerical simulations with a glacial mass balance model, and (4) source of origin of BC aerosols over the HKH region. The hypsometry of nine glaciers taken under study (refer to Table 1 and Section 2.4) is shown in Figure 1(c). The selected glaciers are widely distributed over the HKH region and located near the zone of high $BC_c$ values (discussed in Section 3.2). The out-

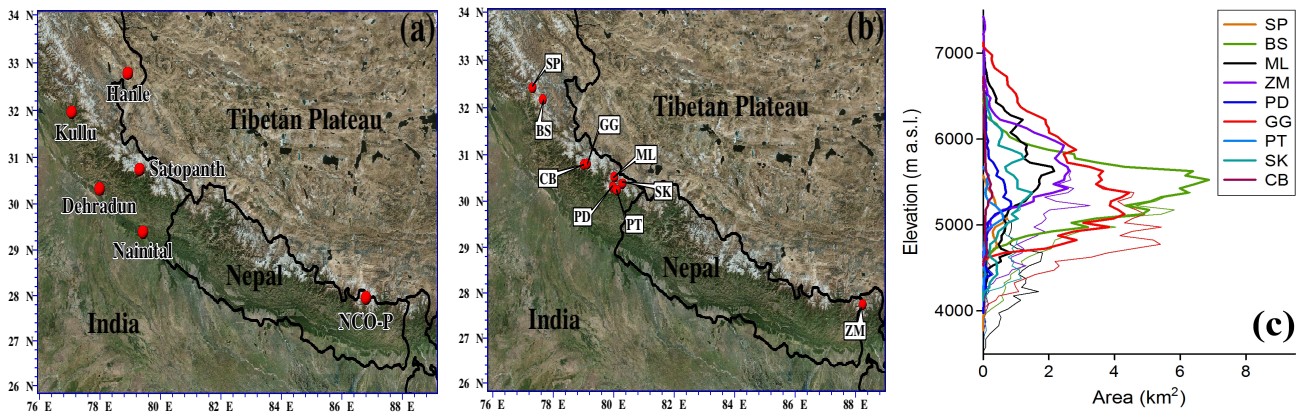

**Figure 1.** Map of the study domain showing (a) six stations of available measurements and (b) simulated glaciers (c) Hypsometry of nine glaciers. Outline of glaciers are based on GAMDAM Glacier Inventory (Nuimura et al., 2015). Thick and thin lines denote debris-free and debris-covered areas respectively. The debris-covered area was excluded from the calculation. Glacier abbreviations and details are listed in Table:1.

line of glaciers are based on Glacier Area Mapping for Discharge from the Asian Mountains (GAMDAM) Glacier Inventory (Nuimura et al., 2015).

## 2   Methodology

### 2.1   Free running BC simulations in GCM

We evaluate BC surface concentration estimated from three $freesimu$ performed with the general circulation model of Laboratoire de Météorologie Dynamique (LMD-ZT GCM) (Hourdin and Armengaud, 1999; Li, 1999; Hourdin et al., 2006) and Spectral Radiation Transport Model for Aerosol Species (SPRINTARS) (Takemura, 2012). The two $freesimu$ with the LMD-ZT GCM comprises of (i) LMD-ZT GCM - with India emissions (GCM-indemiss), (ii) LMD-ZT GCM coupled to the Interactions between Aerosols and Chemistry (INCA) model (LMDORINCA)– with global emissions, or namely GCM-INCA. Aerosol

simulations in GCM are also referred as free running aerosol simulations ($freesimu$) since simulated aerosol fields are not constrained by observations, unlike constrained simulations (described in Section 2.2). We provide only a very brief description of these models here and detailed information could be obtained from references provided for models. Specific description of aerosol treatment and atmospheric transport for GCM-indemiss (GCM with India emissions) is given by Boucher and Pham (2002); Reddy et al. (2004); Verma et al. (2008), for LMDORINCA by Schulz et al. (2006), and for SPRINTARS by Takemura

(2012). GCM-indemiss uses BC emissions over India from Reddy and Venkataraman (2002a, b), over Asia from Streets et al. (2003a), and global emissions of BC are from Cooke et al. (1999); Cooke and Wilson (1996). BC simulation with GCM-indemiss is carried out with a zoom factor of four and three applied in longitude and latitude respectively, having the zoom centralized at 75°E and 15°N, and extending from 50°E to 100°E in longitude and from 5°S to 35°N in latitude; leading to a

resolution of 1° in longitude and 0.8° in latitude over the zoomed region. GCM-INCA global emission inventory for BC is from Generoso et al. (2003) and Bond et al. (2004). BC simulation with GCM-INCA is done in non-zoom mode at a horizontal resolution of 3.75° in longitude and 2.5° in latitude. In both GCM model variants 19 vertical layers with a hybrid sigma-pressure coordinate are used, having five layers below 600 hPa and nine layers above 250 hPa. Model meteorological fields are nudged to 6-hourly data from European Centre for Medium-Range Weather Forecast (ECMWF) meteorological analysis data with a relaxation time of 0.1 day for the year of 2001 in GCM-indemiss, for 2006 in GCM-INCA.

In SPRINTARS, global emission for BC is adopted from the standard inventories for the Coupled Model Inter-comparison Project Phase 5 (CMIP5) (Lamarque et al., 2010; Moss et al., 2010). BC simulation with SPRINTARS is done at a horizontal resolution of approximately $2.8° \times 2.8°$ with 20 vertical layers based on sigma-pressure levels (Takemura, 2012). These simulations are available for a long term period from 1851 to 2010. GCM-indemiss simulation for the year 2001 is compared with SPRINTARS output for 2001 (referred to as SP1). Long-term BC simulation output from SPRINTARS is used to compare model values (referred to as SP2) with observations corresponding to the year of observation. Also, these simulations are used to predict the annual rate of increase in BC concentration over the HKH glaciers as presented in Section 3. The one-sigma uncertainty in mean model simulated atmospheric BC burden based on evaluation in sixteen global aerosol models has been estimated as 42% (Textor et al., 2006).

Among the three $freesimu$ estimates, the prediction of the GCM-indemiss is found in more sanguinity with the measured than that of the rest other models (please refer to Section 3.1). The $freesimu$ estimates from the GCM-indemiss are, therefore, used in $constrsimu$ approach (refer to Section 2.2). The $freesimu$ of GCM-indemiss has also the highest conformity with the measurements at HA stations, hence we utilize BC concentration simulated in GCM-indemiss to estimate BC concentration in snow and BC-induced SAR and their impact over the HKH region (refer to Sections 2.3 and 3.2).

In order to examine sources of BC aerosol over the HKH region due to emissions from near-by and far-off regions and that from various source sectors (e.g., residential biofuel use (BF), open burning of biomass (OB), and fossil fuel (FF) combustion), region- and source-tagged simulations carried out in GCM-indemiss (Verma et al., 2011, 2008) are evaluated. The sectors for the BF source include wood and crop-waste for residential cooking and heating, for OB include forest biomass and agricultural residues, and that for the FF source are coal-fired electric utilities, diesel transport, brick kilns, industrial, transportation, and domestic (Reddy and Venkataraman, 2002a, b).Two sets of experiment were carried out, where aerosols were either tagged by source regions (region–tagged simulation) or source sectors (source–tagged simulation). In the region-tagged BC simulations, the BC aerosol transport and atmospheric processes are simulated for each geographical region with the emissions outside that region being switched off. These source regions are the following: (1) Indo-Gangetic plain (IGP), (2) central India (CNI), (3) south India (SI), (4) northwest India (NWI), (5) southeast Asia (SEA), (6) east Asia (EA), (7) Africa-west Asia (AFWA), and (8) rest of the world (ROW). In the source-tagged BC simulations, the BC aerosol transport and atmospheric processes are simulated for each of the source sector – BF, FF, and OB.

Recently, Wang et al. (2014) introduced an explicit aerosol tagging technique, implemented in the Community Atmosphere Model (CAM5), in which BC emitted from fourteen independent source regions was tagged and explicitly tracked to quantify source-region-resolved characteristics of BC. This tagging technique has been applied to evaluate the source of origin of black

carbon (BC) over the region of Arctic and that over the Himalayas and Tibetan plateau (HTP) (Wang et al., 2014; Zhang et al., 2015). The tagging technique in CAM5 appears similar to that applied for the region-tagged simulation in GCM-indemiss (Verma et al., 2007, 2008, 2011), though, the classified regions of interest in CAM5 and GCM-indemiss are different. The source regions implemented in GCM-indemiss zoom grid were classified on the basis of differences in composition of their

aerosol emission fluxes and their proximity to the Indian Ocean and the subcontinent (Verma et al., 2007). The masked regions on the GCM zoom grid is shown as Figure s1 in the supplementary material provided with this manuscript. The tagged region of "South Asia" (SAS) in Zhang et al. (2015) includes most of the part of the Indian subcontinent (including together the tagged regions, IGP, CNI, SI, NWI in GCM-indemiss). The tagged region of "AFWA" in GCM-indemiss includes the combined tagged regions of Sub-Saharan Africa (SAF), North Africa (NAF), and middle east (MDE) in Zhang et al. (2015).

**Table 2.** Details of BC simulation used in this study

| Experiment | Aerosol Simulations in GCM (For LA, HA & Glaciers) | | |
| | Emissions | Resolution (mode) | Simulation year |
|---|---|---|---|
| GCM-indemiss | India[a], Asia[b], Global[c] | $1° \times 0.8° \times 19$ (zoom) | 2001 |
| GCM-INCA | Global[d] | $3.75° \times 2.5° \times 19$ (non-zoom) | 2006 |
| SPRINTARS | Global[e] | $2.8° \times 2.8° \times 20$ | multi-yr. period |
| Experiment, estimated value[f] | Constrained Aerosol Simulations (For LA) | | |
| | Observational data used | Station Name | Year |
| $Ratio-approach, SC_{bc}^{cr}(i,j,t)$ | $R_{bc}$, ground-truth $\tau_{500}$ | Kullu, DDN, NTL | year of measurement[h] |
| $Profile-approach, SC_{bc}^{cp}(i,j,t)$ | CALIPSO profile ($\sigma$) [g] | Kullu, DDN, NTL | 2008-09 |

[a] Reddy and Venkataraman (2002a, b); [b] Streets et al. (2003a); [c] Olivier and Berdowski (2001); Cooke et al. (2002); Cooke and Wilson (1996); [d] Bond et al. (2004); Generoso et al. (2003); Olivier and Berdowski (2001); [e] Takemura et al. (2000, 2009); Guenther and et al. (1995); Spiro et al. (1992); [f] Estimated value of aerosol species concentration from constrained simulation compared with observation; [g] Profile of $\sigma$ to $\tau$ at 532 nm from CALIPSO; [h] Observed ground-truth $\tau_{500}$ from available studies NTL 2005-12, Kullu 2009-12 and DDN 2007-10;

## 2.2   Constrained BC simulation

The constrained simulation ($constrsimu$) approach takes into account the influence of a possible discrepancy in base emissions, the localized effect of high emission flux (from the plains near LA), and the combined effect of inter-annual variability in meteorological effects (Kumar et al., 2018). This is used to establish an alternate approach for estimating the atmospheric BC concentration by surpassing the error induced specifically due to emissions in source regions which prevail in case of the

free running aerosol simulations. In constrained simulations, GCM-indemiss AOD is constrained by the observed AOD. An inversion algorithm is formulated to obtain the surface concentration of BC from constrained AOD of aerosol constituents. It may be notable that constrained simulations are considered only for the LA stations. The HA stations which are far away from the source of emissions, the impact of regional emission bias is postulated to be minor and aerosol pollutants are primarily governed by the atmospheric transport processes (simulated atmospheric residence time) from the surrounding regions. Our

postulate is supported by the fact that BC from $freesimu$ of GCM-indemiss that are underestimated by a large factor at LA stations (refer Section 3.1 and Table:4), match consistently well with available observations at HA stations (refer to section:3.1

and figure:2). Hence, in the present study, simulated BC from GCM-indemiss is considered a suitable choice (based on analysis of performance of model with measurements, discussed in Section 3.1) for evaluation of BC-induced SAR and its impact on annual snowmelt runoff from the glaciers under study.

The two constrained simulations formulated using the ratio approach (RA) and profile approach (PA) are compared with measurements. On a fundamental basis the two simulations namely, RA and PA, use BC-AOD ($\tau_{bc}^m$) and total AOD ($\tau^m$) from the $freesimu$ of GCM-indemiss to obtain the ratio $R_{bc}$ as follows:

$$R_{bc}(i,j,\bar{t}) = \frac{\tau_{bc}^m(i,j,\bar{t})}{\sum \tau^m(i,j,\bar{t})} \tag{1}$$

where $\bar{t}$ specifies the seasonal mean for winter and pre-monsoon. The $i$ and $j$ corresponds $to$ the grid positions on the spatial map with the resolution of 1° x 0.8° as in GCM-indemiss simulation output. The total AOD from GCM-indemiss, $\tau^m$ is constrained with the ground-based observed AOD ($\tau^o$) which is then scaled with $R_{bc}$ to calculate the constrained BC-AOD ($\tau_{bc}^c$) as given in equation 2.

$$\tau_{bc}^c(i,j,\bar{t}) = R_{bc}(i,j,\bar{t}) \times \tau^o(i,j,\bar{t}) \tag{2}$$

In order to incorporate the $\tau_{bc}^c$ for calculating the RA constrained surface concentration of BC namely, $SC_{bc}^{cr}$ a ratio ($\eta_{bc}$) between the GCM-indemiss BC concentration ($SC_{bc}^m$) and $\tau_{bc}^m$ is considered.

$$\eta_{bc}(i,j,\bar{t}) = \frac{SC_{bc}^m(i,j,\bar{t})}{\tau_{bc}^c(i,j,\bar{t})} \tag{3}$$

The spatial distribution of $\eta_{bc}$ is confined to the same resolution as the GCM-indemiss. Using this $\eta_{bc}$ the constrained surface concentration is calculated based on the theory that the sensitivity of $\tau_{bc}^c$ to $SC_{bc}^{cr}$ is numerically equivalent to the sensitivity of $\tau_{bc}^m$ to $SC_{bc}^m$.

$$SC_{bc}^{cr}(i,j,\bar{t}) = \eta_{bc}(i,j,\bar{t}) \times \tau_{bc}^c(i,j,\bar{t}) \tag{4}$$

The constrained surface concentration of BC as obtained above by RA is evaluated with measurements.

In the PA, station height specific aerosol extinction coefficients for the corresponding grid coordinates $(i,j)$ are taken into account. This involves the calculation of the monthly mean of the aerosol extinction coefficient ($\sigma$) vertical profile derived from Cloud Aerosol Lidar and Infrared Pathfinder Satellite Observations (CALIPSO). The extinction coefficients are available layer wise at a wavelength of 532 nm and vertical resolution $dz$ = 0.06 km. The previously obtained $\tau_{bc}^c$ is disaggregated vertically using the retrieved profile of the aerosol extinction coefficient ($\sigma$ in km$^{-1}$) from CALIPSO renormalized by the CALIPSO AOD, $\tau^{calip}$ averaged during the seasonal periods (as indicated by $\bar{t}$) at given HKH LA stations to obtain the BC extinction

coefficient:

$$\sigma_{bc}(i,j,z,\bar{t}) = \tau_{bc}^c(i,j,\bar{t}) \times \frac{\sigma(i,j,z,\bar{t})}{\tau^{calip}(i,j,\bar{t})} \tag{5}$$

The vertical distribution of the BC concentration, termed as PA constrained concentration ($SC_{bc}^{cp}$), is then calculated based on the formula given below:

$$SC_{bc}^{cp}(i,j,z,\bar{t}) = \frac{\sigma_{bc}(i,j,z,\bar{t})}{\alpha_{bc}(i,j) \times dz} \tag{6}$$

where $\alpha_{bc}$ is the mass extinction coefficient for BC, taken as 10–16 m$^2$g$^{-1}$ (Ram et al., 2010). The uncertainty in the BC surface concentration from $constrsimu$ using RA, taking into account the uncertainty in observed AOD, $\alpha_{bc}$, and inter-annual variability in AOD, is estimated to be within 30% (Kumar et al., 2018). This using PA, through application of CALIPSO data is estimated as within 45%.

**2.3 Estimation of BC concentration in snow and its impact on the annual snowmelt runoff for the glaciers under study**

BC concentration in the snow ($BC_c$, $\mu$g kg$^{-1}$) during pre-monsoon and winter is calculated with the following equation:

$$BC_{c,(ij\bar{t})} = \frac{SC_{bc,(ij\bar{t})}^m \times v_d \times t}{\rho_{ij\bar{t}} \times d} \tag{7}$$

where $SC_{bc,\bar{t}}^m$ is the seasonal mean for atmospheric BC mass concentration ($\mu$g m$^{-3}$) simulated with GCM-indemiss, $v_d$ is
the dry deposition velocity (m s$^{-1}$), $t$ is the season specific time interval considered for snow deposition, $\rho_{\bar{t}}$ is the seasonal mean snow density (kg m$^{-3}$) and $d$ is the snow depth (m). $BC_c$ is calculated assuming a uniform distribution of BC in top 2 cm of pure snow, as it is more contaminated and contributes to a larger albedo reduction than the deeper layers (Tanikawa et al., 2009). Deposition velocity of particles has been inferred being higher than 0.010 cm s$^{-1}$ over land (Nho-Kim et al., 2004); the minimal $v_d$ of 0.010 cm s$^{-1}$ is taken in the present study (similar to Yasunari et al. (2010) for the Himalayan glacier), and a
higher $v_d$ than the above will lead to increase in estimated $BC_c$.

The uncertainty in the estimated $BC_c$ is 69% due to model uncertainty in $SC_{bc}^m$ and lack of information on $v_d$ (model uncertainty for BC dry deposition velocity is 55% consistent with that mentioned in Yasunari et al. (2013) as 0.01–0.054 cm s$^{-1}$). The percentage one-sigma variability in the pre-monsoon mean of $\rho_{ij\bar{t}}$ and $SC_{bc,(ij\bar{t})}^m$ is respectively 26%–30% and 42%–55%. This of $BC_c$ is estimated as 50%–70%.

Deposition of atmospheric BC over the snow also takes place by wet deposition through below cloud scavenging during snow fall events. Calculation of this deposition therefore requires information about size-resolved snow and BC. There is a lack of this information for the Himalayan region from observations and therefore is estimated using numerical models through parametrization of snow scavenging (Ming et al., 2008). Measurements of snowfall from meteorological stations are

not available at and around Hanle (Nair et al., 2013) or that over the HKH stations under study. Using information from Nair et al. (2013) on the rate of wet scavenging based on atmospheric BC measurements and that on snow accumulation depth at Hanle, and snow density from ECMWF, our estimated value of $BC_c$ from wet deposition is 36 $\mu$g kg$^{-1}$ for the pre-monsoon (using the prescribed snow density of 195 kg m$^{-3}$, the value is the same as that in Nair et al. (2013)). This value is obtained in the range of 32–90 $\mu$g kg$^{-1}$ for the HKH glaciers extrapolating the information at Hanle for the entire HKH region and glaciers under study. The total precipitation amount and the precipitation events have been inferred to be notably low during the pre-monsoon season over the HKH region (Bonasoni et al., 2010b; Yasunari et al., 2010). Hence, due to the lack of adequate estimation of wet deposition, calculation of BC impacts on SAR during the pre-monsoon is done neglecting the wet deposition and considering the dry deposition as reasonably the governing mechanism for the removal of atmospheric BC during the period of study.

The seasonal mean of spatial distribution of snow density obtained from ECMWF is used to calculate $BC_c$. This calculation is also done assuming the fixed snow density of 195 kg m$^{-3}$ (for fresh snow) as per information from Yasunari et al. (2010) in order to facilitate comparison between estimation of $BC_c$ from the present study and previous studies over Himalayan stations (e.g. from Nair et al. (2013)).

The percentage reduction in snow albedo ($\alpha_{dec}^{empirical}$)(%) for the estimated BC concentration in snow cover ($\mu$g kg$^{-1}$) is assessed using the following empirical relationship proposed by Ming et al. (2009).

$$\alpha_{dec}^{empirical} = 0.075 \times BC_{c,\bar{t}} + 0.0575 \tag{8}$$

The equation describing the relationship between BC concentration in snow and albedo reduction (Ming et al., 2009) was obtained based on snowpack observations and model calculations. As obtained from Table 4 of Yasunari et al. (2010), the BC-induced SAR over the NCOP from this equation was within $-17\%$ to 8% of that estimated for internal mixture of BC with new snow by Hansen and Nazarenko (2004).

BC-induced SAR is also calculated from the online radiative model, Snow, Ice, and Aerosol Radiation (SNICAR) ($\alpha_{dec}^{SNICAR}$) (Flanner et al., 2007) to compare with the value obtained from equation (8). This calculation is done taking the difference of snow-albedo between two sets of simulation with SNICAR, one with $BC_c$ and the other without $BC_c$. SNICAR uses parameters such as snow grain size, solar zenith angle, species concentration, albedo of the underlying ground, snowpack thickness and snowpack density (refer to Table 3). The effective radius of snow grain size is taken as 100 $\mu$m (Warren and Wiscombe, 1980) and snowpack density and snowpack thickness is referred from ECMWF gridded archive files corresponding to the period of present study.

The relative percentage uncertainty in estimated SAR (%) due to uncertainty in BC$_c$ is estimated as within 30%. The percentage one-sigma variability in SAR associated with the variability in $\rho_{ij\bar{t}}$ is 9%–14%. The BC-induced SAR (%) for each of the corresponding HKH stations is calculated. When compared, it is observed that calculated SAR values from SNICAR are lower than that using the empirical approach (refer to Figure:2(e)).

Lower values from SNICAR than empirical approach is attributed to consideration of external mixing of snow with BC,

and possibly a large snow grain size (of 100 $\mu m$) for addressing snow aging; snow grain radius size of 100 $\mu m$ to 1000 $\mu m$ is considered for new and aged snow by Warren and Wiscombe (1980).

The consideration of old-aged snow may lead to an enhanced BC accumulation onto the snow surface and a higher reduction of snow albedo than estimated in the present study taking into account BC accumulation in the top 2 cm of pure snow (not contaminated from the pre-existing debris cover). Also, there is a possibility of enrichment of BC in snow (not taken into account in the present study) due to the predominance of sublimation compared to precipitation during the period under study, including likely a higher $v_d$ than in the present study which justifies our assumption that we are using a lower bound estimate of SAR from what could be the actual scenario. Also, BC-induced SAR used in the sensitivity analysis of annual snowmelt runoff for a glacier (discussed in the next Section) is considered for $BC_c$ with dry deposition only (due to lack of adequate estimation of wet deposition). All the above would thereby lead to an enhanced $BC_c$ and impact compared to that estimated in the present study.

The present study aims to evaluate the estimated impact of BC aerosols over the glaciers in the HKH region and identify the glacial region most vulnerable to BC-induced impacts considering the lower bound estimates of their concentration in snow and the corresponding BC-induced SAR.

**Table 3.** Input parameters for SNICAR-Online model

| Parameters | Values | reference/source |
|---|---|---|
| Solar zenith angle | $51.8° - 54.7°$ | NOAA |
| Snow grain effective radius | $100 \ \mu m$ | Warren and Wiscombe (1980) |
| Snowpack thickness (m) | $\tau_{ij\bar{t}}$ | ECMWF |
| Snowpack density ($kg \ m^{-1}$) | $\rho_{ij\bar{t}}$ | ECMWF |
| Albedo of underlying ground | Visible: 0.2 $\mu m$ | |
| | Near-Infrared: 0.4 $\mu m$ | ECMWF |
| BC concentration in the snowpack ($\mu g \ kg^{-1}$) | $BC_{cij\bar{t}}$ | calculated from GCM-indemiss simulations |

NOAA: National Oceanic and Atmospheric Administration Solar position calculator;
$\tau_{ij\bar{t}}$ = glacier wise seasonal mean value of snowpack thickness;
$\rho_{ij\bar{t}}$ = glacier wise seasonal mean value of snowpack density;
$BC_{cij\bar{t}}$ = glacier wise seasonal mean value of $BC_c$

## 2.4 Impact of BC-induced SAR on increase in annual snowmelt runoff for glacier

SAR is considered as one of the precursors of excess snow melting. To quantify the impact of BC concentration in the snowpack on annual snowmelt runoff for the glacier (also termed as annual glacier runoff in the text), we calculated the mass balance and runoff from nine selected glaciers with an energy and mass balance model for glaciers (Fujita and Ageta, 2000) (Table:1 and Figure:1(b)). As mentioned earlier, the selected glaciers are widely distributed over the HKH region and located near the zone of high $BC_c$ values (discussed in Section 3.2). We have used a calculation scheme, in which the albedo of glacier surface was reduced from the albedo value corresponding to null value of $BC_c$ during the control run to an albedo value of 0.5, by which we assumed the glacier surface darkened by BC, at a given date and then snowmelt runoff in the following summer season was calculated (Fujita, 2007). Surface albedo of the control run was not fixed value but calculated from snow density so that

it changed temporally and spatially. In the calculation, other meteorological parameters were not changed so that we evaluate the impact of surface darkening by BC only. The lower portions of the selected glaciers are covered by debris mantle. The sub-debris ice melt is highly altered with the debris thickness, and the albedo lowering by BC should not affect the ice melting under the debris layer. We, therefore, excluded the debris-covered area (Figure 1(c)), which is delineated manually by following previous studies in the eastern Himalaya (Nagai et al., 2016; Ojha et al., 2016, 2017). The calculation was performed with the ERA-Interim reanalysis data (Dee et al., 2011), but precipitation was calibrated to set an initial climatic condition for each glacier, which surely affects glacier response to climate change (Sakai et al., 2015; Sakai and Fujita, 2017). In the calculation, Fujita (2007) has pointed out that the timing of surface darkening affected the consequence of annual runoff significantly. Darkened surface during winter could be covered by succeeding snowfall while that in the early melting season significantly enhanced glacier melting and thus runoff. On the other hand, impact of surface darkening in the late melting season was limited because of the remaining short period for melting. Therefore, in the calculation of this study, the date of albedo lowering was changed at 5 day intervals from October to April. Summer mean albedo and annual runoff were calculated , and then averaged for 35 years (1979–2014). The entire process is carried out for each of the glaciers. Output data set is then further analyzed statistically and discussed in Sections 3.2.1 and 3.2.2. The results of annual glacier runoff depth and summer mean albedo so obtained is then used to generate the data set for values of SAR and the corresponding annual runoff increase (ARI), estimating the respective values with-respect-to the control run for each of the nine glaciers. The BC-induced SAR values estimated for each of the glaciers under study is then interpolated with the above data set to estimate the corresponding range for ARI, which is analyzed in Section 3.2.1.

## 3    Results & discussion

### 3.1    Comparison between model estimated and measured BC concentration

Model estimates (free running simulations and constrained simulations as given in Table 2 and 3) are compared to measurements. This comparison is presented as a scatter plot in Figure 2(a)–(d) for HA and LA stations. The dashed line on both the sides of the scatter plot represents a deviation factor of 2 and 0.5 respectively. A detailed performance of the model outputs with respect to the measurements is also provided in Table 4.

### 3.1.1    Lower altitude Stations

At LA stations, estimated values of BC concentration from $freesimu$ of GCM-INCA and SPRINTARS are underestimated by a large factor, being 6 to 23 times the measured values; with that from $freesimu$ of GCM-indemiss being 2 to 11 times (refer to Table 4) and hence exhibits a better performance than that of the rest other models. At LA stations, we also evaluate the estimates from $constrsimu$ (obtained by ratio approach, RA; and profile approach, PA; refer to Section 2.2) with measurements. The comparison between model estimates ($freesimu$ and $constrsimu$) and measurements is presented as a scatter plot in Figure 2(b)–(c). Estimates of BC concentration from $constrsimu$ (Figure 2(c)) exhibit a better concurrence than that

from $freesimu$ (Figure 2(b)) with the measured concentration at all the three LA stations. Among the $constrsimu$ (RA and PA approaches), the RA estimates deviate most of the times by a factor of two compared to the measurements (as shown by the dashed lines on both sides of the equivalent line, Figure 2(c)), amounting to about 30%–100% of the measured values, with the normalized bias being $-70\%$ to $-35\%$ except for Nainital where the pre-monsoon RA values mirror the measurements.

Compared to RA, the PA estimates exhibit a better coherence with the measured values, with these estimates amounting to 90-100% of measured data and a normalized bias of $-14\%$ to 13%. It is seen that the absolute bias for PA estimates is within their estimated uncertainty of as large as 45% (as discussed in Section 2.2). A better conformity of PA estimates with the measured data also emphasizes the consideration of vertical profile of BC aerosols and comparison of model values near approximate height corresponding to the field measurements. Also, a better performance of $constrsimu$ compared to $freesimu$ estimates

authenticates our postulate that inappropriate estimates of emission in the $freesimu$ is the primary reason for anomalous prediction by the models at LA stations which are in proximity to emission sources.

### 3.1.2   Higher altitude stations

The modeled values of simulated BC concentration from $freesimu$ at HA stations (relatively remotely located and mostly influenced by transport of aerosols) (Figure 2(a)) have more agreement to the field measurements than that at LA stations

(Figure 2(b)). It may be noted that HA stations are accompanied with relatively lower values (0.08 $\mu$g m$^{-3}$ to 0.4 $\mu$g m$^{-3}$) of measured BC concentration compared to LA stations (1.1 $\mu$g m$^{-3}$ to 6.5 $\mu$g m$^{-3}$).

While estimates from GCM-indemiss has a normalized bias (NB) range of –6 to 30 % at HA stations for both pre-monsoon and winter seasons. Whereas those from GCM-INCA exhibits a normalized bias range of –17 to 17 % except at NCOP where the overestimation by the model is as high as 2–8 times for both the seasons. SPRINTARS simulations for versions SP1 (year

of simulation 2001) and SP2 (year corresponding to the year of measurement) show a general negative bias range of –21% to –70% except at Hanle where SPRINTARS exhibits a positive bias of 30% to 120%. GCM-indemiss values are hence further analyzed by comparing it with SPRINTARS (SP1 simulation). GCM-indemiss values are 3–4 times that of the SPRINTARS values for all the stations in both seasons except at Hanle where the predicted concentrations by both are similar. The variability in annual mean of BC concentration (2000–2010) simulated from SPRINTARS is less than 1% and that for years corresponding

to measurement is 4%–8%; this is consistent with the inter-annual variability estimated in measured BC concentration which is 7%–9%. The long-term SPRINTARS simulations for BC concentration are used to estimate the rate of increase (w.r.t 1961) in BC concentration and thereby in BC concentration in snow ($BC_c$).

Among the three $freesimu$ estimates, especially that of GCM-indemiss (which is also at a higher spatial resolution than rest other models) has the highest conformity with the measurements at HA stations. Also, the NMB of $freesimu$ from

GCM-indemiss for HA is within the estimated one-sigma uncertainty in mean model simulated atmospheric BC burden. It may be noted that the feature of a relatively lower magnitude and spatial variability of measured BC concentration over HA stations compared to that over LA stations, is also seen in $freesimu$ of GCM-indemiss estimates. The estimated atmospheric BC concentration over the HKH glaciers is as low as 0.251–0.253 $\mu$g m$^{-3}$ over the Bara Shigri (northern Himalaya)/Zemmu (eastern Himalaya) glacier to as large as 0.517–0.532 $\mu$g m$^{-3}$ over the Chorabari/Gangotri (northern Himalaya) glacier. Hence,

our analysis of the model estimates indicate that $freesimu$ estimates of GCM-indemiss used to evaluate BC distribution for HA stations and HKH glaciers, represent reasonably consistently the relative degree of transport of BC and thereby the feature of spatial and seasonal BC distribution across the HKH sites, including the HKH glaciers. Although, to reduce further the NMB of GCM-indemiss estimates with-respect-to the respective observations over the HKH stations and to obtain a more accurate

magnitude of BC concentration over the HKH glaciers, it is required to examine BC transport simulation in the $freesimu$ of the chemical transport model (CTM) with a better resolved fine grid scale transport processes and vertical distribution. This requirement is indeed justified as the improved prediction of $constrsimu$ is estimated for BC concentration from the PA compared to that from the RA, with the consideration of vertical profile of BC aerosols and comparison of model values near approximate height corresponding to the field measurements at LA stations (please refer to Figure 2c, Table 4, and Section

3.1.1). The setting up of a simulation experiment with a better spatially and vertically resolved BC transport processes with the $freesimu$ of a CTM for the HKH region is under progress, with an implementation also of the improved and the latest BC emissions over the Indian region as an input into the model. Results from this simulation will be presented in a future study. Thus, discrepancies between model and measurements at HA stations is likely attributed to uncertainties which stem out from instrument error, analytical errors, detection capability (pertaining to measurements of low values of atmospheric

BC concentration), including degree of accuracy of model processes governing the atmospheric residence time of BC aerosols over HA stations (pertaining to $freesimu$ simulations). In summary, our analysis shows that model estimates ($freesimu$ of GCM-indemiss over HA stations and HKH glaciers and $constrsimu$ over LA stations) with the estimated uncertainty (as large as 45%) represent adequately, in general, the feature and magnitude of BC distribution over the HKH sites. As discussed previously, the $freesimu$ of GCM-indemiss has the highest conformity with the measurements at HA stations, hence we

utilize BC concentration simulated in GCM-indemiss to estimate BC concentration in snow and BC-induced SAR and their impact over the HKH region.

## 3.2    Analysis of spatial distribution of BC concentration in snow: estimates of impact on SAR and annual glacier runoff

The comparison of BC concentration in snow ($BC_c$) between GCM-indemiss derived values and that obtained from available studies (e.g. Nair et al. (2013); Yasunari et al. (2010)) at HKH HA stations is presented in Figure 2(d). As mentioned earlier, snow density used to estimate $BC_c$ is extracted from the ECMWF ERA-Interim daily reanalysis data set. In order to compare our $BC_c$ estimates with that from available study, we also estimate $BC_c$ using the fixed snow density of 195 kg m$^{-3}$ as done in the available studies. These estimates using fixed snow density and that using ECMWF data at HA stations are presented in

Figure:2(d).

The estimated value of $BC_c$ from the present study compares well with that obtained from previous work, and has a variation of 10% to 27% from the earlier studies. These estimates compare relatively well at Hanle and NCOP, however, at Satopanth during winter they are overestimated compared to the respective measured value. The percentage difference of $BC_c$ estimates using fixed snow density with respect to that using ECMWF retrieved values is found to be 34% − 38%. The estimated BC$_c$

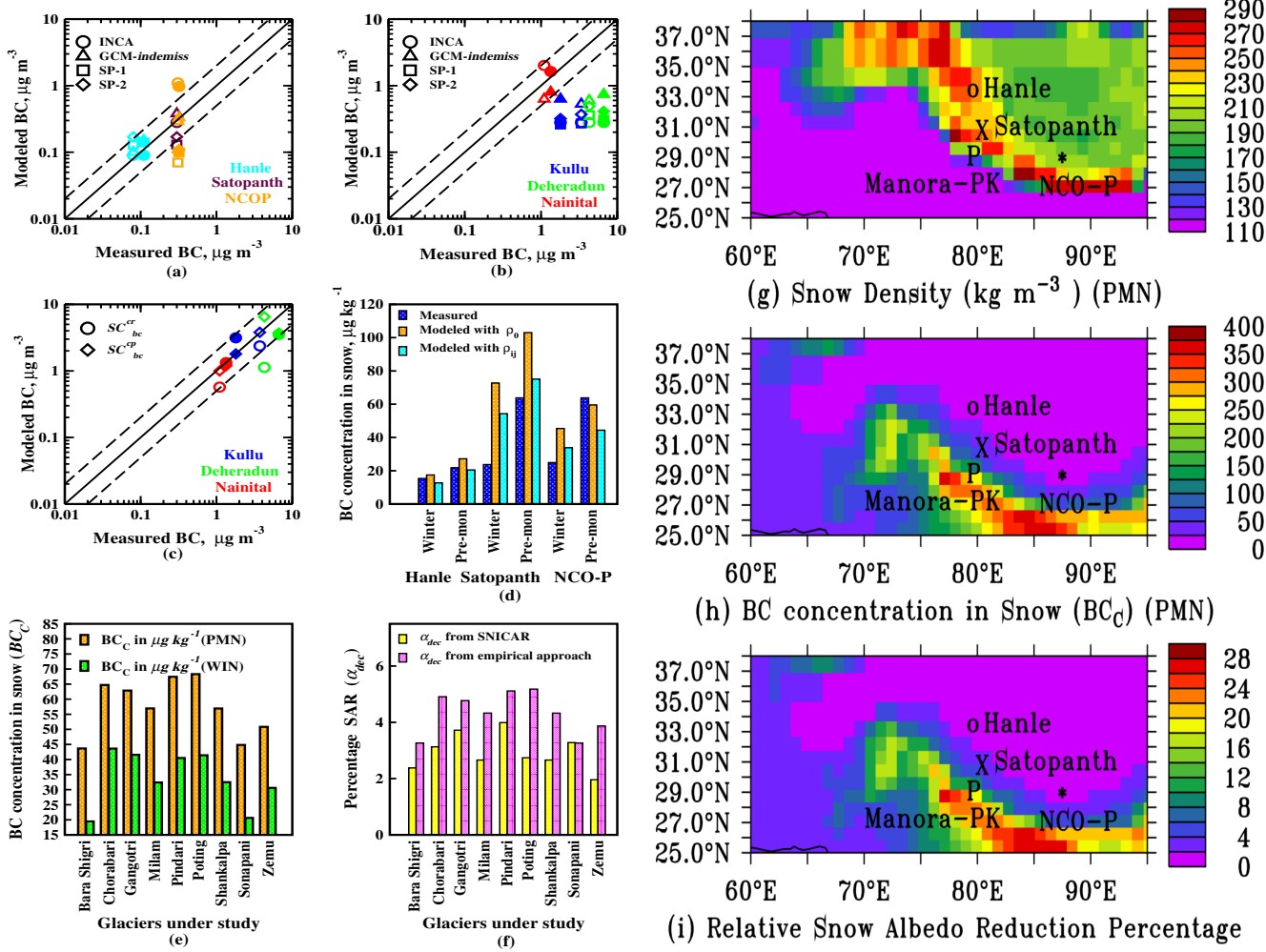

**Figure 2.** Figures showing spatio-seasonal variation of atmospheric mass BC concentration for pre-monsoon (filled symbols) and winter (unfilled symbols) for (a) Higher altitude, (b) lower altitude stations (c) Modeled versus measured atmospheric BC concentration at lower altitudes using RA and PA, (d) Station wise BC concentration in snow ($BC_c$) and comparison of modeled and measured values for Higher altitudes, (e) Glacier wise value of $BC_c$ considering only dry deposition for winter and pre-monsoon season, (f) Glacier wise percentage snow albedo reduction (SAR) from both SNICAR and empirical approach for pre-monsoon; Spatial distribution of (g) averaged snow density from ECMWF for pre-monsoon season (PMN), (h) average of the modeled $BC_c$ for PMN and (i) percentage SAR estimated using empirical approach ($\alpha_{dec}^{empirical}$) for PMN

(with fresh snow consideration) over HKH glaciers (refer to Figure 2e) during the pre-monsoon (20–68 $\mu$g kg$^{-1}$) and that during the month of June (84 $\mu$g kg$^{-1}$) in the present study is inferred to be comparable to that measured over the southeastern Tibetan plateau e.g. for fresh snow/ice sample (four nos. during June) at Demula glacier (29.26°N, 97.02°E; 56.6±26.1 $\mu$g kg$^{-1}$) (Zhang et al., 2017). This inference is consistent with the observational studies indicating the BC concentration in snow over the southeastern Tibetan plateau is the most comparable to those over the Himalayan regions (Zhang et al., 2017). We

**Table 4.** Comparison of model output range of atmospheric BC concentration with the observed values at different stations

| | HA | | | LA | | |
|---|---|---|---|---|---|---|
| | Hanle | Satopanth | NCO-P | Kullu | Dehradun | Nainital |
| **Model simulation** | | | | | | |
| | **NMB**[1] **win**[3]**(pmn**[4]**)** | | | **Factor**[2] **win**[3]**(pmn**[4]**)** | | |
| GCM-INCA | 17 (−17) | −7(−30) | 20(3) | 12(6) | 23(15) | 1(1) |
| GCM-indemiss | 30(19) | 27(30) | −6(3) | 6(3) | 11(6) | 2(2) |
| SPRINTARS | 65(30) | −58(−65) | −41(69) | 12(7) | 18(14) | 2(2) |
| | | | | | | |
| **Constrained simulation** | | | | | | |
| | | | | **NMB**[1] **win**[3]**(pmn**[4]**)** | | |
| | | | | **Factor**[2] **win**[3]**(pmn**[4]**)** | | |
| Ratio Approach | | | | 37(55) | 74(47) | 48(1) |
| | | | | 1.6(2.2) | 1.9(1.0) | 3.8(1.9) |
| Profile Approach | | | | −12(−1) | −4(14) | 27(10) |
| | | | | 1.4(1.1) | 0.9(0.9) | 1.0(1.2) |

[1] NMB: Normalized Mean Bias (%) = $\frac{Observed\ BC\ concentration\ value\ from\ literature\ -\ Modeled\ value\ of\ BC\ concentration}{Observed\ BC\ concentration\ value\ from\ literature} \times 100$

[2] Factor = $\frac{Observed\ BC\ concentration\ value\ from\ literature\ for\ LA\ stations}{Modeled\ value\ of\ BC\ concentration\ at\ respective\ stations}$

[3] win = Monthly average value of December, January and February

[4] pmn = Monthly average value of March, April and May

also compare the estimated $BC_c$ from our study with the observed value at a few other stations over the Himalayas (e.g. East Rongbuk glacier, Kangwure, Qiangyong). This comparison over the mentioned stations has also been carried out in a study by Kopacz et al. (2011) using estimates from Geos-Chem simulations. The estimated $BC_c$ from the present study at East Rongbuk glacier (28°N, 88°E) for the month of October (Ming et al., 2009), at Kangwure (28.5°N, 85.8°E), and Qiangyong (28.3°N, 90.3°E) for the month of July (Xu et al., 2006) are respectively 13, 17, and 24 $\mu g\ kg^{-1}$, which are found to be lower than the respective observed value by 27%, 23%, and 44%. Some of the above discrepancies are expected due to the comparison of the model estimates, which are monthly averaged (corresponding to the month of observation), with the respective measured values at stations which are mostly based on a single sample observation (Ming et al., 2009; Xu et al., 2006). The above bias is, however, within the range of uncertainty in $BC_c$ as estimated in the present study. It is also noted that the bias in the estimated $BC_c$ for the above stations as obtained from the Geos-Chem simulations (Kopacz et al., 2011) is respectively −155%, −22% (negative bias indicates the model value is higher than the observed), and 54%, which is found to be higher than that from the present study. The lower estimated values of $BC_c$ than the measured, specifically at Qiangyong, are, although, as expected in our study due to non-consideration of wet deposition in the estimation of $BC_c$. Nevertheless, the comparable values of the model estimates (within the range of uncertainty in $BC_c$) with the respective observation indicate the conformity with our consideration of the dry deposition as a governing removal process of atmospheric BC during the period of study over the region.

We further examine the estimates of $BC_c$ and the relative percentage SAR (%) for the HKH glaciers under study as presented in Figures 2(e) and 2(f) respectively. The pre-monsoon mean of $BC_c$ and BC-induced SAR for the HKH glaciers is higher by 39%–76% than the winter mean. Among the glaciers, Pindari, Poting, Chorabari, and Gangotri, located over the northern HKH region near the Manora peak exhibit specifically high value of $BC_c$ (63–68 $\mu$g kg$^{-1}$) and BC-induced SAR ($\approx$5%).

We also evaluate the percentage rate of increase in $BC_c$ estimates over the HKH glaciers from 1961 to 2010 (with respect to 1961) (Figure 3(a)) estimated from the available long-term simulation of atmospheric BC concentration in SPRINTARS. While there is a relative increase of $BC_c$ from 1961 to 2010 for all glaciers under study, the percentage increase for Pindari glacier is 26% compared to that for Zemmu, Milam and Bara Shigri which being 130%, 55% and 50% respectively.

The decadal increase indicates the possibility of the largest relative impact due to $BC_c$ on BC-induced glacial mass balance
for the Zemmu glacier (over eastern Himalayas), under conditions of similar glacier hypsometry and climatic setting. Nevertheless, the $BC_c$ for Pindari glacier is about 7-times that for the Zemmu glacier and hence would be more vulnerable than Zemmu, to these impacts. A sensitivity analysis of the change in the glacial mass balance due to relative change in hypsometry and climatic setting is discussed in Section 3.2.2.

In order to identify the location of the hot-spots over the HKH region, we also analyse the spatial distribution of pre-monsoon
mean of $BC_c$ (Figure 2(h)) and SAR (Figure 2(i)). The spatial distribution of pre-monsoon mean of snow density from the ECMWF is presented in Figure 2(g). The pre-monsoon mean of snow density is seen in the range 190–290 kg m$^{-3}$ over the HKH glaciers. Please refer to Figure 1(b) for the locations of the glaciers in the study region. High $BC_c$ (40 $\mu$g kg$^{-1}$ to 125 $\mu$g kg$^{-1}$) and the BC-induced SAR (2% to 10%) over the HKH region are located between coordinates 70°E to 90°E and 26°N to 33°N, specifically over the grids around Manora peak. A gradually increasing trend westwards from NCOP and as one moves
north towards Hanle over the snow-covered (Figure 2) HKH region is observed in the spatial distribution of $BC_c$ and SAR. The extent of the glaciers studied under this work (as listed in Figure 1(b)) over the region of 78°E to 88°E and 27°N to 34°N thus allow us to locate the glaciers which are the most affected due to $BC_c$; this is examined in the next Section. The ablation zone of Gangotri glacier (located from the field measurements), which lies at approximately 30°93′ N and 79° E is also found to be within the zone of predicted high $BC_c$ values. High $BC_c$ in the range of 55 $\mu$g kg$^{-1}$ to 100 $\mu$g kg$^{-1}$ in the ablation zone
of Gangotri near the snout, Gomukh, is estimated in the present study.

### 3.2.1   Estimates of impact on annual glacier runoff

The simulated impact of albedo reduction on annual snowmelt runoff (mm of water equivalents per year, mm w.e.y$^{-1}$) from glaciers using a glacial mass balance model (Fujita and Ageta, 2000) is presented in Figure 3(c). Error bars of each point is calculated as the standard deviations of the 35 years simulated period (1979-2014). The gradient of linear regression (Figure
3(c), Table 5), which represents an amount of annual glacier snowmelt runoff increase per unit decrease in albedo, is an indicative of the sensitivity of a glacier to albedo change. This gradient is estimated as $-6834$ to $-9945$ mm w.e.y$^{-1}$ (refer to Table 5) per unit of albedo decrease for HKH glaciers under study. The annual glacier snowmelt runoff increase per unit decrease in albedo for a cold Tibetan glacier has been estimated as about 3670 mm w.e.y$^{-1}$ (Fujita, 2007; Fujita et al., 2007). This indicates the HKH glaciers are about 2–3 times more sensitive to albedo change (i.e. a higher rate of glacier runoff

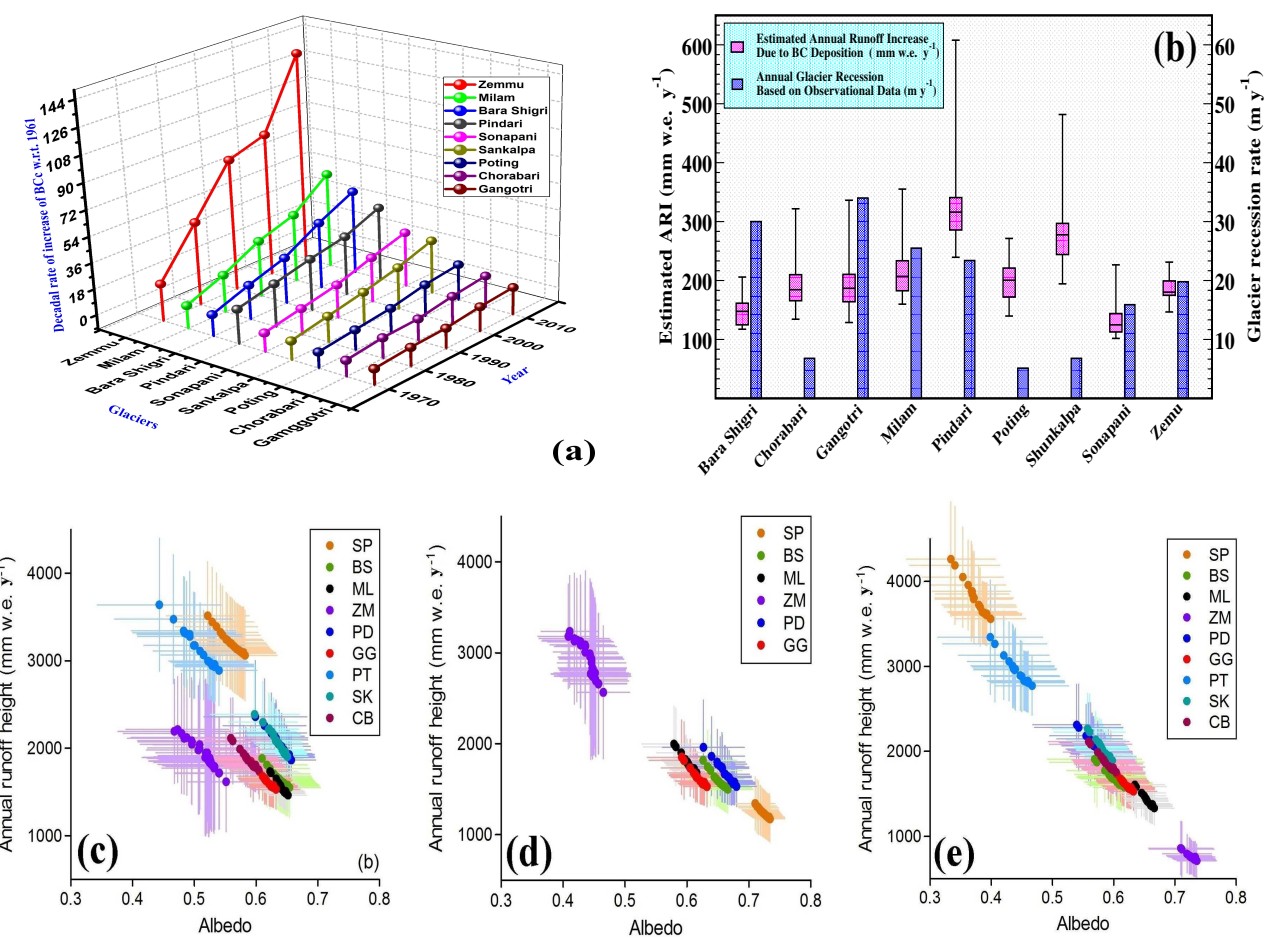

**Figure 3.** (a) Glacier wise decadal trend (1961–2010, % rate of increase w.r.t 1961) of $BC_c$ based on atmospheric BC concentration ($\mu g$ m$^{-3}$) modeled over the glacier using SPRINTARS, (b) Estimated increase in annual glacier runoff (mm w.e.y$^{-1}$) shown as box-whisker plot and average yearly recession (m y$^{-1}$) for the nine glaciers, (c) Simulated summer mean albedo and annual runoff depth of glaciers, and those of sensitivity analysis for (d) a single glacier hypsometry at different ERA-Interim grids and (e) different glacier hypsometries at a single ERA-Interim grid

increase per unit decrease in albedo) than the cold Tibetan glacier. Our derived BC-induced SAR value for each of the HKH glaciers (Figure 2(f)) is then incorporated (Section 2.4) into the results of the numerical simulation to calculate the increase in annual snowmelt runoff from the glaciers which is displayed as box-plot in Figure:3(b). The Figure presents the estimated annual runoff increase (ARI) in mm w.e.y$^{-1}$ for debris-free part of glaciers under study along with glacier wise average annual

5    recession in meters (as obtained from available observational studies, refer to Table 1). The calculated values of ARI due to BC-induced SAR (Figure:3(b)) show the median value of ARI for the HKH glaciers under study being greater than 120 mm w.e.y$^{-1}$, with the highest value (316 mm w.e.y$^{-1}$) for the Pindari glacier and followed by that for the Shunkalpa glacier (278

mm w.e.y$^{-1}$). Although the median ARI for Milam glacier is around 261 mm w.e.y$^{-1}$ (lower than the Pindari), this glacier witnesses the highest percentage increase (18%) of ARI w.r.t the control run (with null value of BC$_c$), followed by Pindari (16% increase). The mean BC-induced ARI across the glaciers is in the range 135–321 mm w.e.y$^{-1}$, with the highest (lowest) being for Pindari (Sonapani). The one-sigma variability in ARI across them due to BC-induced SAR is estimated as 58 mm

w.e.y$^{-1}$. The calculated values of the rate of ARI per unit BC-induced SAR (%) as shown in Table 5, indicate this rate being specifically high ($> 55$ mm w.e.y$^{-1}$ per unit % SAR) for Milam, Pindari, and Shunkalpa glacier. From the Figure, it can also be seen that the Gangotri glacier is the most rapidly shrinking one among the HKH glaciers under study, having maximum yearly recession of as high as 34 meters, (Sangewar and Kulkarni, 2011), followed by Bara Shigri, Milam, and Pindari. Five out of the total nine glaciers are retreating at a rate higher than 20 meters per year (range varying from 6.8 to 34 m y$^{-1}$), which

is really alarming.

The percentage uncertainty in albedo due to the impact of surface darkening by BC associated with the different climatic conditions is 5%–20% and in the corresponding estimated annual snowmelt runoff is 13%–47% (Figure:3c). The uncertainty in the corresponding increase in annual snowmelt runoff (Figure:3b) is 10%–23%.

In summary, the present study leads to identify glaciers over the HKH region being vulnerable to BC-induced impacts,

which includes those affected by a large value of $BC_c$ and BC-induced SAR (e.g. Pindari, Poting, Chorabari, and Gangotri), in addition to those being specifically highly sensitive to BC-induced impacts (e.g. Shunkalpa, Pindari, Milam, and Zemmu). The Pindari glacier feeds the river Pindar which runs for 124 km with a catchment area of 1688 km before it's confluence with river Alaknanda at Karnaprayag. It is highly essential that the ARI from this glacier be quantified to manage the abundant meltwater. Our analyses show that there is an enhanced $BC_c$ in the present decade as compared to the earlier decades due to the enhanced

BC concentrations originating from anthropogenic activities.

### 3.2.2   Sensitivity analysis of the annual glacier runoff–albedo relationship

The estimated impact of $BC_c$ on annual glacier runoff is affected not only by the amount of deposited BC but also by glacier hypsometry and climatic setting. To quantify these effects, we further calculated the sensitivity of annual glacier runoff to albedo change with two different configurations: (1) with a single glacier hypsometry at different ERA-Interim grids, and (2)

with different glacier hypsometries at a single ERA-Interim grid. The configuration (1) demonstrates how different climatic settings affect the annual runoff sensitivity (Fig:3(d)) while configuration (2) shows how different altitudinal profiles affect the results (Fig:3(e)). We use the hypsometry of Gangotri Glacier (GG) because of its wide range in altitude (Fig:1(c)). Configuration (1) resulted in similar annual runoff heights (Fig:3(d)) and annual runoff sensitivity (GG hypsometry, Table:5) for the neighboring grids (ML and PD) except for two grids (BS and ZM). Zemmu Glacier (ZM) is located in the eastern Himalaya

and thus situated higher elevation than the other glaciers (Fig:1(c)). If the GG hypsometry is applied to the ZM grid, the large ablation area should produce more meltwater under warmer condition. Albedo reduction should enhance the ice melting in such a way, warmer the condition, more will be the runoff. This is why the annual runoff sensitivity is so negative. On the other hand, Sonapani Glacier (SP) is the northwestern margin of the studies domain. Larger accumulation area of the GG glacier than the SP glacier should reduce the area-averaged annual runoff because the albedo reduction should not affect glacier melting at

high altitude. Configuration (2) demonstrates how glacier hypsometry affect the amount and sensitivity of annual runoff (GG grid, Table:5 and Fig:3(e)). Neighboring glaciers (ML, PD, GG, PT, SK, and CB) show similar results to the control calculation (Fig:3(c)) while glaciers from different climate show contrasting results. The ZM glacier is situated at higher altitude than the GG glacier (Fig:1(c)), the amount and sensitivity of annual runoff are significantly reduced, and vice versa for the SP

glacier (Fig:3(e) and Table:5). Spatial contrast of climate and glacier hypsometry affect how these glaciers respond to surface darkening by BC.

**Table 5.** Sensitivity of annual runoff and runoff increase (ARI) to albedo change for the glaciers considered in this study, along with sensitivity of annual runoff to albedo due to change in climatic condition and glacier hypsometry

| Glacier Name (abbreviation) | Sensitivity of annual runoff to albedo [1] (mm w.e. $y^{-1}$ albedo$^{-1}$) | Sensitivity of ARI to SAR [2] (mm w.e. SAR$^{-1}$ | Sensitivity of annual runoff to albedo for GG hypsometry [3] (mm w.e. $y^{-1}$ albedo$^{-1}$) | Sensitivity of annual runoff to albedo for GG grid [4] (mm w.e. $y^{-1}$ albedo$^{-1}$) |
|---|---|---|---|---|
| Bara Shigri (BS) | -6841 | 44.751 | -7263 | -6500 |
| Chorabari (CB) | -7935 | 47.925 | Same as GG | -7935 |
| Gangotri (GG) | -7700 | 48.684 | -7700 | -7700 |
| Milam (ML) | -9586 | 62.465 | -8554 | -8700 |
| Pindari (PD) | -9031 | 59.341 | -8198 | -8093 |
| Poting (PT) | -7971 | 43.041 | Same as PD | -8116 |
| Shunkalpa (SK) | -9945 | 64.713 | Same as PD | -9357 |
| Sonapani (SP) | -6834 | 39.787 | -6371 | -11084 |
| Zemmu (ZM) | -7039 | 38.818 | -12738 | -5301 |

[1] Slope of annual runoff (mm w.e.$y^{-1}$) to albedo from Figure:3(c); [2] Slope of increase in annual runoff (mm w.e.) to BC induced percentage SAR; [3] Slope of annual runoff (mm w.e.$y^{-1}$) to albedo from Figure:3(d), where Gangotri (GG) hypsometry implies that the calculation with the same hypsometry of Gangotri Glacier (GG), as a sensitivity test of different climatic condition by using the single hypsometry; [4] Slope of annual runoff (mm w.e.$y^{-1}$) to albedo from Figure:3(e), where Gangotri (GG) grid implies that the calculation with the same ERA-Interim grid including Gangotri Glacier (GG), as a sensitivity test of different hypsometry by using the single ERA-Interim grid

We believe the heterogeneity in ARI estimated across the glaciers to be reasonably adequate. This is because of the application of the consistent estimates of atmospheric BC from the *freesimu* of GCM-indemiss, which showed the best conformity with observations, and merging these estimates with the relevant information from the available observation over the HKH sites

to assess the impact of BC aerosols over the HKH glaciers. Further, the estimated $BC_c$ from our study (please refer to Section 3.2) has been found to be consistent with the available information on observed values at the stations over the Himalayas (e.g. East Rongbuk glacier, Kangwure, Qiangyong). Furthermore, the BC-induced ARI is estimated based on the "sensitivity of glacier runoff to albedo change" for each of the glaciers under study using glacier energy-mass balance model. The present study evaluated the estimated impact of BC aerosols over the glaciers in the HKH region and identified the glacial region most

vulnerable to BC-induced impacts considering the lower bound estimates of their concentration in snow and the corresponding BC-induced SAR. Though, it is believed that the feature of the spatial distribution of BC-induced impacts as estimated from the integrated modelling approach in the present study is reasonably consistent. It is, however, required to examine a more

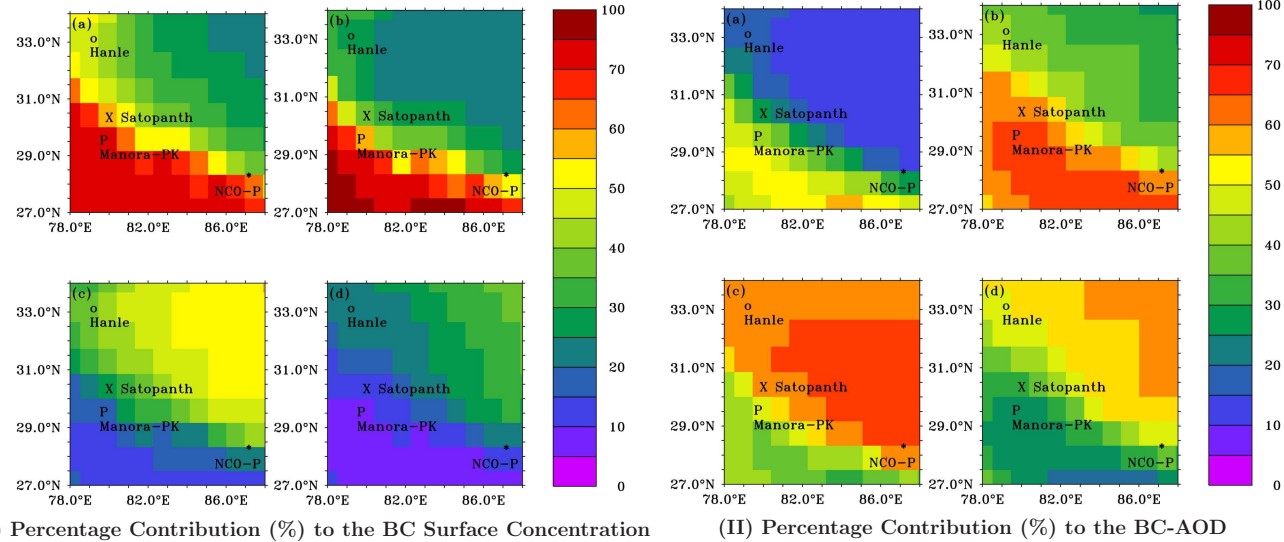

**Figure 4.** Spatial distribution of mean of (I) surface BC concentration and (II) BC-AOD for winter (left panel) and pre-monsoon (right panel) due to relative percentage contribution (%) of emissions from the IGP (top panels) and AFWA (bottom panels).

accurate magnitude of these impacts on the individual glaciers with the improved estimates (reduced bias with observations) of atmospheric BC concentration, in conjunction with the improved representation of aerosol processes and parametrization applied for the HKH sites in the model. This improvement is mainly concerned with the aided information required from more detailed and frequent aerosol observations of crucial aerosol parameters including the dry and wet deposition rates, mixing of BC in snow, in addition to that of meteorology, e.g. snowpack density and height, precipitation, snow grain size, surface albedo, over a widely distributed stations across the HKH sites. The uncertainty in estimated BC concentration in snow, which further propagates in estimated SAR and ARI, is mainly due to the lack of information on aerosol processes for the HKH sites. The prevailing uncertainty on atmosphere–cryosphere processes and hindrance in obtaining a more robust estimate of the aerosol impact on Himalayan glaciers due to too sparse and short-term observations has also been inferred in previous studies (e.g. Qian et al. (2015); Ménégoz et al. (2014); Matt et al. (2018)). It is also, therefore, necessary to carry out a more detailed validation of model estimates for BC aerosols alongwith the meteorology with a widely distributed observations across the HKH sites.

### 3.3 Source of origin of BC aerosols over HKH region

#### 3.3.1 Contribution of BC emissions from near-by and far-off regions to BC aerosols

The relative distribution (%) of BC surface concentration (Figure 4(I)) and BC-AOD (Figure 4(II)) due to BC emissions originating from near-by region, IGP (BC concentration or BC-AOD due to emissions from the IGP to the total BC concentration or BC-AOD) and far-off region, Africa-west Asia (AFWA) during winter and pre-monsoon seasons over the HKH region is es-

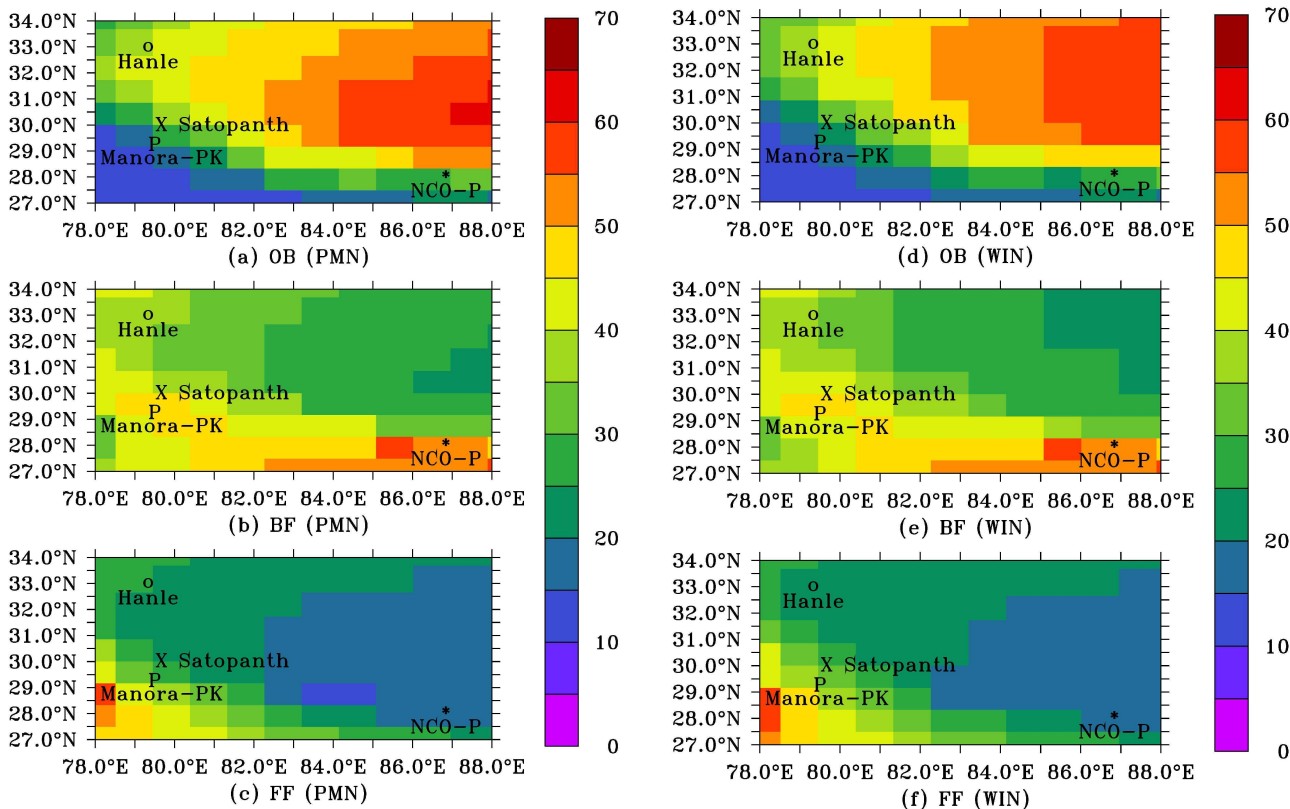

**Figure 5.** Spatial distribution of (left panel, a-c) pre-monsoon, and (right panel, d-f) winter mean of surface BC concentration due to relative percentage contribution (%) of BC emissions from open burning (OB), biofuel (BF) and fossil fuel (FF) combustion.

timated. These estimations are done from region-tagged simulations in the GCM over the HKH region (please refer to Section 2.1). The spatial distribution (%) of winter and pre-monsoon mean of surface BC concentration (BC-AOD) due to emissions from the IGP is 70% to 90% (45% to 80%) of the total surface BC concentration (BC-AOD) over the part of the HKH region south to 30°N. This distribution decreases as we move towards the north (north to 30°N), at higher latitudes in the Himalayan

5   region. The spatial distribution (%) of surface BC concentration (BC-AOD) due to emissions from AFWA increases towards the northern grids (north to 30°N), with these being 25% to 40% (50% to 60%) in winter and 15% to 25% (40% to 55%) in pre-monsoon. It is found that while BC concentration and BC-AOD due to emissions from the IGP has a slightly higher value during pre-monsoon than winter, it is vice-versa for that due to emissions from AFWA. This feature is inferred likely due to emissions of biomass burning (crop-waste and forest fires) which is more prominent during pre-monsoon over the IGP but that

10   during winter over AFWA as obtained from the information of firecounts data from satellite-based measurements (Pohl et al., 2014; Verma et al., 2014). Estimates of pre-monsoon and winter mean of surface BC concentration due to relative contribution (%) of emissions from OB, BF and FF combustion from source-tagged (refer to Section 2.1) BC simulations in LMDZT-GCM are also presented respectively in Figures 5(a to c) and 5(d to f). These Figures indicate that while BC surface concentration

during winter and pre-monsoon are influenced mostly due to biofuel emissions over part of the HKH region south to 30°N (where there is a relatively higher contribution from emissions over the IGP), these are due to biomass burning over north to 30°N, thus corroborating the above inference. The high $BC_c$ and BC-induced SAR over glaciers near to the Manora peak, over northern Himalayan region are thus found to be mainly due to contribution of BF emissions from the IGP region.

The present study is in agreement with the inference based on source-tagging technique implemented in CAM5 (as also mentioned in Section 2.1, Methodology) and that using adjoint of the Geos-Chem model (Zhang et al., 2015; Kopacz et al., 2011), on the predominant contribution from biomass burning (BB) emissions to BC aerosols over the Himalayas and Tibetan Plateau (HTP). Although, unlike the previous studies (e.g. Kopacz et al. (2011); Zhang et al. (2015)), the present study provides a more specific domain over the HKH region, e.g. south to 30°N (covering the region of analysis of BC aerosols for Himalayas in Zhang et al. (2015)), where a predominant contribution of BC emissions originating in IGP is evinced, corroborating that originating in SAS by Zhang et al. (2015). An enhanced contribution of emissions from Sub-saharan Africa region over the Himalayas and central plateau during winter compared to that during pre-monsoon as inferred in Zhang et al. (2015) is also consistent with that in the present study. Further, for the region north to 30°N within the HKH region under study (covering the region of analysis of BC aerosols for the central plateau in Zhang et al. (2015)), where the present study indicates a predominating contribution to BC aerosols from open burning (OB) emissions and that originating mainly in AFWA, is in disagreement with Zhang et al. (2015), which once again indicates that from BB emissions originating mainly in SAS. This disagreement is expected due to (i) tagged region of "SAS" in Zhang et al. (2015) also includes parts of west Asia, unlike that of a more specific domain over the Indian subcontinent "IGP" in the present study, (ii) the source of BC is considered from the combined source sector of BF and OB as BB in Zhang et al. (2015), unlike that in the present study. While BC emissions from the BF combustion are predominant over the IGP region (part of SAS from Zhang et al. (2015)), these from OB are over Africa region specifically during the winter season. It is, however, noted that a considerable contribution of BC emissions from Africa region over the Himalayas (Mount Everest), and this being larger than that from the Indian region during the winter month (January) compared to that during the pre-monsoon month (April) has been inferred based on the source analysis using adjoint of the Geos-Chem model (Kopacz et al., 2011); this inference is in corroboration with the present study.

## 4 Conclusion

We evaluated the impact of BC aerosols and their sources of origin over the HKH region and identified the glaciers most sensitive to such impacts. This is done through utilizing BC concentration estimated from the free running ($freesimu$) BC simulations with the LMDZT-GCM and SPRINTARS and that from constrained simulation ($constrsimu$) approach. Sensitivity analysis of BC-induced snow-albedo reduction (SAR) to increase in annual glacier runoff is done through numerical simulations with glacial mass balance model for identified nine glaciers.

While $freesimu$ of GCM-indemiss exhibited a relatively good comparison with available observations over the high-altitude locations, the $constrsimu$ exhibited that with over the low-altitude (LA) locations. A good comparison of $constrsimu$ indicated the discrepancy in emissions in $freesimu$ being a primary reason for the anomalous performance of $freesimu$ over

the LA stations. Estimates of BC concentration in snow ($BC_c$) was consistent with that obtained from available study, and its spatial mapping led to identify the hot-spot zone over the HKH region. Analysis of $BC_c$ over the identified glaciers located in the vicinity of the hot-spot zone over the HKH region showed specifically high $BC_c$ and BC-induced SAR over glaciers near to the Manora peak, over northern Himalayan region (e.g., Pindari, Poting, Chorabari, and Gangotri), with their highest being that for Pindari and Poting. Using long-term BC simulations from SPRINTARS, the relative (%) rate of increase of $BC_c$ for the present years (with respect to 1961) was the highest (130%) for the glacier (Zemmu) near eastern Himalayan region, and was five times that for the Pindari glacier; thereby indicating the largest relative impact of the BC-induced glacial mass balance for the Zemmu glacier among glaciers studied, under conditions of similar glacier hypsometry and climatic setting.

Sensitivity analysis of annual glacier runoff to BC-induced albedo change indicated that HKH glaciers are more sensitive than cold Tibetan glaciers. The median of annual glacier runoff increase due to BC-induced SAR was estimated to be greater than 120 mm w.e.y$^{-1}$ for HKH glaciers with the highest being for Pindari. Among the HKH glaciers, the rate of increase in annual glacier runoff per unit BC-induced percentage SAR was specifically high for Shunkalpa, Pindari, and Milam (over northern Himalayas). The present study provided information on specific glaciers over the HKH region as identified being vulnerable to BC-induced impacts (affected by high BC-induced SAR in addition to that being sensitive to BC-induced impacts). This information about pollution-induced impact on specific glaciers is also seen to be consistent with that on signatures of observed glacial recession indicating a relatively higher recession for Gangotri, Milam, and Pindari among the HKH glaciers.

While the source of BC aerosols over the HKH region south to 30°N (including the hot-spot zone) was inferred being primarily from the biofuel emissions originating in the nearby region (IGP), it was over north to 30°N from open burning emissions originating in far-off region (AFWA).

Information on BC-induced impacts on SAR and sensitivity of annual glacier runoff over the HKH region will be utilized in a future study to understand the pollution-induced hydrological changes downstream and also to predict the fate of rivers originating from the region in the long run.

It is targeted to further assess the degree of improvement in the prediction of the magnitude of BC aerosols over the HKH sites through setting up BC transport simulation with a better resolved fine grid scale transport processes and vertical distribution, including also an implementation of the improved and the latest BC emissions over the Indian region as an input into the model. Results from this simulation with their implication on HKH glaciers will be presented in a future study.

## 5  Acknowledgments

This work was supported through a grant received from the Department of Science and Technology (INT/NOR/RCN/P05/2013) and from the Ministry of Environment, Forest, and Climate Change (14/10/2014-CC (Vol.II)), Govt. of India. We acknowledge accomplishment of GCM simulations through computing time provided by the Institut du Développement et des Ressources en Informatique Scientifique (IDRIS) of the CNRS, France.

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
