# Peer review of "Simulations of black carbon (BC) aerosol impact over Hindu-Kush Himalayan sites: validation, sources, and implications on glacier runoff"

_Atmospheric Chemistry and Physics, 2018_

## Referee Comment (RC1) · Anonymous Referee #2 · 6 Nov 2018

**ACP-2018-869 review comments**

The present study tries to characterize BC source attributions and its implications on glacier runoff over the Hindu Kush Himalayan region. The motivation of the work is established in the introduction. Satellite data, black carbon (BC) observations, reanalysis data, and global model are used and subsequently introduced with select relevant details. In the main part of the work, the results are presented and analyzed, and the authors carefully quantify and discuss the model performance, especially the implication of BC from surrounding regions to glacier runoff in different seasons.

The manuscript is thorough, clear, compelling, and presents the results with good figures and tables. I recommend publication after attending to the following major and minor revisions.

1, There is a good review paper including discussions of BC effects over the Himalayas and Tibetan Plateau (Qian et al., 2015). We suggest the author refer it in the section of introduction.

Qian, Y., Yasunari, T. J., Doherty, S. J., Flanner, M. G., Lau, W. K., Ming, J., Wang, H., Wang, M., Warren, S. G., and Zhang, R.: Light-absorbing particles in snow and ice: Measurement and modeling of climatic and hydrological impact, Adv. Atmos. Sci., 32, 64-91, 2015.

2, The present work evaluates the model performance about BC concentrations in air, but lack of evaluations for BC-in-snow

concentrations. Please add, because the author also thinks the absorbing effect of BC in snow is important in this region. There are some observations of BC concentration and radiative forcing in snow across the Tibetan Plateau and Himalayas, such as Zhang et al., (2017).

Zhang, Y., et al. (2017), Light-absorbing impurities enhance glacier albedo reduction in the southeastern Tibetan plateau, J. Geophys. Res. Atmos., 122, doi:10.1002/2016JD026397.

3, As the author acknowledged, the wet deposition is an important process for BC simulations in the model, so modeled precipitation evaluations against the meteorological station observations or reanalysis data are essential for confidence.

4, In present study, an aerosol tagging method was used, please provide a little bit more details about this technique in your model in the section of methodology. Recently, an explicit emission tagging method has been developed in a global-aerosol model (Wang et al., 2014), could the author add some descriptions about the differences between two tagging methods?

Wang, H., Rasch, P. J., Easter, R. C., Singh, B., Zhang, R., Ma, P. L., Qian, Y., and Beagley, N.: Using an explicit emission tagging method in global modeling of source-receptor relationships for black carbon in the Arctic: Variations, Sources and Transport pathways, J. Geophys. Res.-Atmos., 119, 12888-12909, doi:10.1002/2014JD022297, 2014.

5, In fact, previous works pay attention to characterize BC source-receptor relationships over the Tibetan Plateau and Himalayas. The author should describe the same points, differences and the new discoveries with previous works, such as Kopacz et al., (2011) and Zhang et al., (2015).

Kopacz, M., et al. (2011), Origin and radiative forcing of black carbon transported to the Himalayas and Tibetan Plateau, Atmos. Chem. Phys., 11, 2837–2852, doi:10.5194/acp-11-2837-2011, 2011.

Zhang, R., et al. (2015), Quantifying sources, transport, deposition, and radiative forcing of black carbon over the Himalayas and Tibetan Plateau, Atmos. Chem. Phys., 15, 6205-6223, https://doi.org/10.5194/acp-15-6205-2015, 2015.

---

## Referee Comment (RC2) · Anonymous Referee #1 · 8 Nov 2018

The study estimates the implication of black carbon (BC) deposition on annual snow melting of some representative Himalayan glaciers (Table 1). It uses GCM modelled values of BC in atmosphere to estimate corresponding BC concentration in snow. Subsequently, associated snow albedo reduction (SAR) values using SNICAR offline model is calculated over these glaciers. While these approaches and estimates are not entirely new, two interesting parts of the study are the increasing trend in BC concentration in snow over these glaciers and the estimates of the annual runoff increase due to BC deposition. Largely, the manuscript is well organized and logically analysed, but,

needs clarification/addition at various places.

It is worth publication after addressing the following major concerns:

1) Only SPRINTARS is used beyond Figure 2 in the study mainly because it provides the longest model dataset. Therefore, It sometimes becomes confusing about the purpose of GCM-INCA and GCM-indemiss in the study. The authors could mainly focus on evaluation of SPRINTARS output (collocated with observations) and proceed with further analysis. Moreover, GCMs are for different years. Please clarify. The authors may consider to shorten Section 2.1.

2) The methodology leading to ARI calculation (Section 2.4) should be elaborated. For atmospheric researchers not expert with glacier mass balance presently many things are not clear. For example,

a) Two annual runs are made for each glacier with Glacier MB model; One with ambient albedo simulated by ERA-interim and the other reduced to 0.5 for all the days in a year for each of the 35 years separately?

b) What is the ambient albedo values used in MB model and Why is it reduced to 0.5 particularly (i.e. not something else like 0.75 ?). Realistic snow albedo over Glaciers are ~0.8 during early summer. If it is reduced to 0.5, then a reduction of 0.3 (~30%) is simulated which is manifold higher than the SAR estimates of ~5 %. Thus, please clarify with example the calculations.

c) What is the X-axis in figure 3C? Is it summer mean albedo or reduction in albedo in the mass balance model (confused with Line 23 page 14). Also include in caption clearly.d) Elaborate more on how ARI is calculated using SAR estimates (Line 9 page 10).

3) The spatial resolution of the models used in the study is relatively very coarse compared to the spatial extend of the glaciers which affects the life cycle of BC estimates due to misrepresentation in terrain and surface emissions. This can contribute a lot to

the differences seen in the observation and model values as well as favors the better performance of PA approach than RA approach because of inclusion of vertical profiles. Please include appropriate discussions in relevant places (like ∼ Page 11 Line 5).

4)A major concern is that the study uses coarse model simulations and offline approaches to estimate SAR, but, eventually uses it to compare and contrast the ARI at very fine spatial (glacier) level. In this view, the quantitative differences reported in this study in estimates of ARI, SAR, BC between the glaciers may be very uncertain. The authors have mentioned about the uncertainties in various steps used in this study for example uncertainties in BC in atmosphere estimates (45%), BC in snow estimates (50-70%) and SAR estimates (20-30%). In a sense, all these processes contribute serially to calculation of increase in snowmelt runoff. Moreover, there is uncertainties involved in the process of calculating snowmelt runoff increase from glacier mass balance model also. At the same time, the variability in ARI of various glaciers is within 150 mm w.e. y-1 (Figure 3b), which is ∼50% of the mean ARI over Himalayas. Please include a discussion on these issues towards end of the study.

---

## Author Comment (AC1) · 13 Jan 2019

**Response to Reviews of manuscript entitled "Simulations of black carbon (BC) aerosol impact over Hindu-Kush Himalayan sites: validation, sources, and implications on glacier runoff (acp-2018-869)"**

We thank the Editor and Reviewers for their valuable comments, suggestions and corrections.

Specific changes made in response to the comments are described below. Along with the response, we have also attached a "manuscript document" which shows the change introduced for the revised manuscript. The blue-colored text in the attached document indicate the specific changes included in the main text of the manuscript.

**Response to Reviewers' comments**

**Reviewer 2:**

The present study tries to characterize BC source attributions and its implications on glacier runoff over the Hindu Kush Himalayan region. The motivation of the work is established in the introduction. Satellite data, black carbon (BC) observations, reanalysis data, and global model are used and subsequently introduced with select relevant details. In the main part of the work, the results are presented and analyzed, and the authors carefully quantify and discuss the model performance, especially the implication of BC from surrounding regions to glacier runoff in different seasons.

The manuscript is thorough, clear, compelling, and presents the results with good figures and tables. I recommend publication after attending to the following major and minor revisions.

1. There is a good review paper including discussions of BC effects over the Himalayas and Tibetan Plateau (Qian et al., 2015). We suggest the author refer it in the section of introduction. Qian, Y., Yasunari, T. J., Doherty, S. J., Flanner, M. G., Lau, W. K., Ming,

J., Wang, H., Wang, M., Warren, S. G., and Zhang, R.: Light-absorbing particles in snow and ice: Measurement and modeling of climatic and hydrological impact, Adv. Atmos. Sci., 32, 64-91, 2015.

Response: Thanks for the comment. This is done. We have referred the suggested paper in Introduction. We have also referred to this paper while discussing on uncertainty aspects on page 20 of the attached revised manuscript. Please see blue lines in the attached revised manuscript with the response for the changes.

2. The present work evaluates the model performance about BC concentrations in air, but lack of evaluations for BC-in-snow concentrations. Please add, because the author also thinks the absorbing effect of BC in snow is important in this region. There are some observations of BC concentration and radiative forcing in snow across the Tibetan Plateau and Himalayas, such as Zhang et al., (2017).

Zhang, Y., et al. (2017), Light-absorbing impurities enhance glacier albedo reduction in the southeastern Tibetan plateau, J. Geophys. Res. Atmos., 122, doi:10.1002/2016JD026397.

Response: Thanks for the suggested reference. We have now included more comparisons of BC concentration in snow over HKH sites with the available information over the HKH region. Please refer to blue lines in Section 3.2 for the changes in the revised manuscript attached with the Response.

Comparison of BC concentration in snow from our study with the available information over the HKH region at Hanle, Satopanth, and NCOP from a previous study has already been presented in Figure 2d and discussed in Section 3.2. As per Reviewer's suggestion we have now also reviewed the suggested reference for BC concentration in snow ($BC_c$) observed over the southeastern Tibetan plateau. The estimated $BC_c$ (with fresh snow consideration) over HKH glaciers (refer to Figure 2e in the manuscript) during the pre-monsoon (20–68 $\mu$g kg$^{-1}$) and that during the month of June (84 $\mu$g kg$^{-1}$) in the present study is inferred to be comparable to that measured over the southeastern Tibetan plateau e.g. for fresh snow/ice sample (four nos. during June) at Demula glacier (29.26°N, 97.02°E; 56.6±26.1 $\mu$g kg$^{-1}$) (Zhang et al., 2017). This inference is consistent with the observational studies indicating the BC concentration in snow over the southeastern Tibetan plateau is the most comparable to those over the Himalayan regions (Zhang et al., 2017).

Further, we have now also compared the estimated $BC_c$ from our study with the observed value at a few other stations over the Himalayas (e.g. East Rongbuk glacier, Kangwure, Qiangyong). This comparison over the mentioned stations has also been carried out in a study by Kopacz et al. (2011) using estimates from Geos-Chem simulations. The estimated $BC_c$ in snow from the present study at East Rongbuk glacier (28°N, 88°E) for the month of October (Ming et al., 2009), at Kangwure (28.5°N, 85.8°E), and Qiangyong (28.3°N, 90.3°E) for the month of July (Xu et al., 2006) are respectively 13, 17, and 24 $\mu$g kg$^{-1}$, which are found to be lower than the respective observed value by 27%, 23%, and 44%. Some of the above discrepancies are expected due to comparison of the model estimates, which are monthly averaged (corresponding to the month of observation), with the respective measured value at stations which are mostly based on a single sample observation (Ming et al., 2009; Xu et al., 2006). The above bias is, however, within the range of uncertainty in $BC_c$ as estimated in the present study. It is also noted that the bias in the estimated $BC_c$ for the above stations as obtained from the Geos-Chem simulations (Kopacz et al., 2011) is respectively $-155\%$, $-22\%$ (negative bias indicates the model value is higher than the observed), and 54%, which is found to be higher than that from the present study. The lower estimated values of $BC_c$ than the measured, specifically at Qiangyong, are, although, as expected in our study due to non-consideration of wet deposition in the estimation of $BC_c$. Nevertheless, the comparable values of the model estimates (within the range of uncertainty in $BC_c$) with the respective observation indicate the conformity with our consideration of the dry deposition as a governing removal process of atmospheric BC during the period of study over the region.

3. As the author acknowledged, the wet deposition is an important process for BC simulations in the model, so modeled precipitation evaluations against the meteorological station observations or reanalysis data are essential for confidence.

Response: The wet deposition values of BC mentioned in the manuscript is estimated taking into account the rate of wet scavenging of BC based on atmospheric BC measurements and that on snow accumulation depth at Hanle from Nair et al. (2013), and snow density from ECMWF. Measurements of snowfall from meteorological stations are not available at and around Hanle (Nair et al., 2013) or that over the HKH stations under study. Our estimated value of $BC_c$ from wet deposition is 36 $\mu$g kg$^{-1}$ for the pre-monsoon (using the prescribed snow density of 195 kg m$^{-3}$ at Hanle, the value is the same as that in Nair et al. (2013)). This value is obtained in the range of 32–90 $\mu$g kg$^{-1}$ for the HKH glaciers extrapolating the information at Hanle for the entire HKH region and glaciers under study. The total precipitation amount and the precipitation events have been inferred to be notably low during

the pre-monsoon season over the HKH region (Bonasoni et al., 2010; Yasunari et al., 2010). Hence, due to lack of adequate estimation of wet deposition, calculation of BC impacts on snow albedo reduction (SAR) during the pre-monsoon is done neglecting the wet deposition and considering the dry deposition as reasonably the governing mechanism for the removal of atmospheric BC during the period of study. These informations have already been provided in the manuscript, please refer to blue lines in Section 2.3, page 9, in the revised manuscript attached with the response. As also mentioned in the manuscript, the present study aims to evaluate the estimated impact of BC aerosols over the glaciers in the HKH region and identify the glacial region most vulnerable to BC-induced impacts considering the lower bound estimates, from what could be the actual scenario, of their concentration in snow and the corresponding BC-induced SAR.

4. In present study, an aerosol tagging method was used, please provide a little bit more details about this technique in your model in the section of methodology. Recently, an explicit emission tagging method has been developed in a global-aerosol model (Wang et al., 2014), could the author add some descriptions about the differences between two tagging methods?

Wang, H., Rasch, P. J., Easter, R. C., Singh, B., Zhang, R., Ma, P. L., Qian, Y., and Beagley, N.: Using an explicit emission tagging method in global modeling of source-receptor relationships for black carbon in the Arctic: Variations, Sources and Transport pathways, J. Geophys. Res.-Atmos., 119, 12888-12909, doi:10.1002/2014JD022297, 2014.

Response: Thank you for the suggested reference. We have now included some more details on tagged simulation in the revised version of the manuscript. Please refer to blue lines in Section 2.1, pages 5 and 6, in the revised manuscript attached with the Response. However, to keep the methodology as concise as possible, we have referred to our previous work and provided additional details in the supplementary material also provided now with the manuscript. We have now also referred to the suggested paper by Wang et al. (2014) and included required details in Methodology.

In order to examine sources of BC aerosol over the HKH region due to emissions from near-by and far-off regions and that from various source sectors (e.g., residential biofuel use (BF), open burning of biomass (OB), and fossil fuel (FF) combustion), region- and source-tagged simulations carried out in GCM-indemiss (Verma et al., 2011, 2008) are evaluated. The sectors for the BF source include wood and crop-waste for residential cooking and heating, for OB include forest biomass and agricultural residues, and that for the FF source are

coal-fired electric utilities, diesel transport, brick kilns, industrial, transportation, and domestic (Reddy and Venkataraman, 2002a,b). Two sets of experiments were carried out for the present work, where aerosols were either tagged by source regions (region–tagged simulation) or source sectors (source–tagged simulation). In the region-tagged BC simulations, the BC aerosol transport and atmospheric processes are simulated for each geographical region with the emissions outside that region being switched off. In the source-tagged BC simulations, the BC aerosol transport and atmospheric processes are simulated for each of the source sector – BF, FF, and OB.

Recently, Wang et al. (2014) introduced an explicit aerosol tagging technique, implemented in the Community Atmosphere Model (CAM5), in which BC emitted from fourteen independent source regions was tagged and explicitly tracked to quantify source-region-resolved characteristics of BC. This tagging technique has been applied to evaluate the source of origin of black carbon (BC) over the region of Arctic and that over the Himalayas and Tibetan plateau (HTP) (Wang et al., 2014; Zhang et al., 2015). The tagging technique in CAM5 appears similar to that applied for the region-tagged simulation in GCM-indemiss (Verma et al., 2007, 2008, 2011), though, the classified regions of interest in CAM5 and GCM-indemiss are different. The source regions implemented in GCM-indemiss zoom grid were classified on the basis of differences in composition of their aerosol emission fluxes and their proximity to the Indian Ocean and the subcontinent (Verma et al., 2007). These source regions are the following: (1) Indo-Gangetic plain (IGP), (2) central India (CNI), (3) south India (SI), (4) northwest India (NWI), (5) southeast Asia (SEA), (6) east Asia (EA), (7) Africa-west Asia (AFWA), and (8) rest of the world (ROW). The masked regions on the GCM zoom grid is shown as Figure s1 in the supplementary material provided with this manuscript. The tagged region of "South Asia" (SAS) in Zhang et al. (2015) includes most of the part of the Indian subcontinent (including together the tagged regions, IGP, CNI, SI, NWI in GCM-indemiss). The tagged region of "AFWA" in GCM-indemiss includes the combined tagged regions of Sub-Saharan Africa (SAF), North Africa (NAF), and middle east (MDE) in Zhang et al. (2015).

5. In fact, previous works pay attention to characterize BC source-receptor relationships over the Tibetan Plateau and Himalayas. The author should describe the same points, differences and the new discoveries with previous works, such as Kopacz et al., (2011) and Zhang et al., (2015).
Kopacz, M., et al. (2011), Origin and radiative forcing of black carbon transported to the Himalayas and Tibetan Plateau, Atmos. Chem. Phys., 11, 28372852, doi:10.5194/acp-11-

2837-2011, 2011.

Zhang, R., et al. (2015), Quantifying sources, transport, deposition, and radiative forcing of black carbon over the Himalayas and Tibetan Plateau, Atmos. Chem. Phys., 15, 6205-6223, https://doi.org/10.5194/acp-15-6205-2015, 2015.

Response: Thank you for the suggested papers. We have now referred these in the revised manuscript and discussed our study also in perspective of the suggested works. Please see blue lines in Section 3.3, page 22, for changes in the revised manuscript attached with the response. These are also given as follows:

[revised manuscript text omitted]

. Figure s1: Masked regions on GCM zoom grid representing tagged source regions taken under study; the classification of source regions are as following with different colours indicated in bracket: 1. IGP (light green) 2. CNI (orange), 3. SI (yellow), 4. NWI (red), 5. SEA (green), 6. EA (blue), 7. AFWA (pink), 8. ROW (dark green).

**1   Description of region- and source-tagged simulation**

In order to examine sources of BC aerosol over the HKH region due to emissions from near-by and far-off regions and that from various source sectors (e.g., residential biofuel use, open burning of biomass, and fossil fuel combustion), region- and source-tagged simulations carried out in GCM-indemiss (Verma et al., 2011, 2008) are evaluated. Fig. s1 shows the masked regions on the GCM zoom grid, which include the Indo-Gangetic Plain (IGP), central India (CNI), south India (SI), northwest India (NWI), southeast Asia (SEA), east Asia (EA), Africa-west Asia (AFWA), and rest of the world (ROW). The source regions were classified on the basis of differences in composition of their aerosol emission fluxes and their proximity to the Indian Ocean and the subcontinent (Verma et al., 2007). Out of the parts of Indian region (Indo-Gangetic plain, central India, south India, northwest India), Indo-Gangetic plain has the highest emission flux followed by that of northwest India, central India, and south India. Emission fluxes from the Indo-Gangetic plain, central India, south India are mainly composed of sulfate, organic matter, inorganic matter followed by black carbon. Northwest India emissions are mainly dust followed by sulfate. Southeast Asia emissions are mainly composed of organic matter followed by sulfate. Africa-west Asia emissions are mainly composed of dust and organic matter. Emission flux from east Asia and rest of the world are mainly composed of dust followed by sulfate. In the region-tagged BC simulations, the BC aerosol transport and atmospheric processes are simulated for each geographical region with the emissions outside that region being switched off.

In the source-tagged BC simulations, the BC aerosol transport and atmospheric processes are simulated for each of the source sector – biofuel (BF), fossil fuel (FF), and natural source. The sectors for the BF source include wood and crop-waste for residential cooking and heating, for OB include forest biomass and agricultural residues, and that for the FF source are coal-fired electric utilities, diesel transport, brick kilns, industrial, transportation, and domestic (Reddy and Venkataraman, 2002a, b). The natural source included sulphur from volcanic and biogenic sources, terpenes from the vegetation or natural OM, dust from arid regions, and sea-salt.

---

## Author Comment (AC2) · 13 Jan 2019

**Response to Reviews of manuscript entitled "Simulations of black carbon (BC) aerosol impact over Hindu-Kush Himalayan sites: validation, sources, and implications on glacier runoff (acp-2018-869)"**

We thank the Editor and Reviewers for their valuable comments, suggestions and corrections.

Specific changes made in response to the comments are described below. Along with the response, we have also attached a "manuscript document" which shows the change introduced for the revised manuscript. The blue-colored text in the attached document indicate the specific changes included in the main text of the manuscript.

**Response to Reviewers' comments**

**Reviewer 1:**

The study estimates the implication of black carbon (BC) deposition on annual snow melting of some representative Himalayan glaciers (Table 1). It uses GCM modelled values of BC in atmosphere to estimate corresponding BC concentration in snow. Subsequently, associated snow albedo reduction (SAR) values using SNICAR offline model is calculated over these glaciers. While these approaches and estimates are not entirely new, two interesting parts of the study are the increasing trend in BC concentration in snow over these glaciers and the estimates of the annual runoff increase due to BC deposition. Largely, the manuscript is well organized and logically analysed, but, needs clarification/addition at various places.

It is worth publication after addressing the following major concerns:

1. Only SPRINTARS is used beyond Figure 2 in the study mainly because it provides the longest model dataset. Therefore, It sometimes becomes confusing about the purpose of GCM-INCA and GCM-indemiss in the study. The authors could mainly focus on evaluation

of SPRINTARS output (collocated with observations) and proceed with further analysis. Moreover, GCMs are for different years. Please clarify. The authors may consider to shorten Section 2.1.

Response: In order to spatially map as adequately as possible the estimates of atmospheric BC concentration and BC concentration in snow including the corresponding SAR over the HKH region, an integrated approach merging the relevant information from observations with a relatively consistent atmospheric chemical transport model estimates is applied in the present study.

In the present study, we evaluate BC concentration estimated from the free running ($freesimu$) aerosol simulations using Laboratoire de Météorologie Dynamique atmospheric General Circulation Model (LMDZT-GCM) and Spectral Radiation Transport Model for Aerosol Species (SPRINTARS) over the HKH region. This evaluation includes a comparison of the simulated BC concentration with observations (refer to Sections 3.1 and 3.2), thereby leading to identifying the most consistent $freesimu$ estimates with observations out of the three $freesimu$. The comparison is done with the available observations for winter and pre-monsoon at locations, classified as low-altitude (LA) stations (e.g. Nainital,Kullu, and Dehradun, refer to Figure:1(a)), which are in close proximity to emission sources; and high-altitude (HA) stations (e.g. Hanle, NCO-P, and Satopanth, Figure:1(b)), which are relatively remotely located and mostly influenced by transport of aerosols. Constrained aerosol simulation ($constrsimu$) is also formulated using the simulated aerosol characteristics from the identified $freesimu$, and $constrsimu$ estimates are evaluated over the LA stations.

Among the three $freesimu$ estimates, the prediction of the GCM-indemiss is found in more sanguinity with the measured than that of the rest other models (please refer to Section 3.1). The $freesimu$ estimates from the GCM-indemiss are, therefore, used in $constrsimu$ approach (refer to Section 2.2). The $freesimu$ of GCM-indemiss has also the highest conformity with the measurements at HA stations, hence we utilize BC concentration simulated in GCM-indemiss to estimate BC concentration in snow and BC-induced SAR and their impact over the HKH region (refer to Sections 2.3 and 3.2). The above details are now clearly mentioned in the revised manuscript. Please refer to blue lines in Section 1, page 3, and Section 2.1, page 5 (Lines 16–19) for the changes in the revised manuscript.

We have provided only a very brief description of GCM simulations in Section 2.1 which is required for the methodology of the present study. While presenting this description, we

have tried to keep the Section as concise as possible mentioning the detailed information which could be obtained from references provided for models.

The purpose to analyse the long-term SPRINTARS simulations for BC concentration is mainly to (i) know the variability in annual mean of BC concentration (2000–2010), taking into account the period of simulation year for *freesimu* of GCM-indemiss, GCM-INCA, and of measurements data used, which are for different years, (ii) evaluate the temporal trend of the BC concentration and thereby of BC concentration in snow (BCc) in recent years (w.r.t to that in 1961). GCM-indemiss simulation for the year 2001 is compared with SPRINTARS output for 2001 (referred to as SP1). Long-term BC simulation output from SPRINTARS is used to compare model values (referred to as SP2) with observations corresponding to the year of observation. These comparisons are presented in Section 3.1 of the manuscript. SPRINTARS simulations for versions SP1 (year of simulation 2001) and SP2 (year corresponding to the year of measurement) show a general negative bias range of –21% to –70% except at Hanle where SPRINTARS exhibits a positive bias of 30% to 120%. GCM-indemiss values are hence further analyzed by comparing it with SPRINTARS (SP1 simulation). GCM-indemiss values are 3–4 times that of the SPRINTARS values for all the stations in both seasons except at Hanle where the predicted concentrations by both are similar. The variability in annual mean of BC concentration (2000–2010) simulated from SPRINTARS is less than 1% and that for years corresponding to measurement is 4%–8%; this is consistent with the inter-annual variability estimated in measured BC concentration which is 7%–9%. The uncertainty estimated for model estimates of BC burden takes into account the inter-annual variability. The estimated glacier wise decadal trend (1961–2010, % rate of increase w.r.t 1961) of BC concentration in snow (BCc) as obtained from SPRINTARS is presented in Figure 3a and discussed in Section 3.2.

2. The methodology leading to ARI calculation (Section 2.4) should be elaborated. For atmospheric researchers not expert with glacier mass balance presently many things are not clear. For example,

a. Two annual runs are made for each glacier with Glacier MB model; One with ambient albedo simulated by ERA-interim and the other reduced to 0.5 for all the days in a year for each of the 35 years separately? b. What is the ambient albedo values used in MB model and Why is it reduced to 0.5 particularly (i.e. not something else like 0.75 ?). Realistic snow albedo over Glaciers are ≈0.8 during early summer. If it is reduced to 0.5, then a reduction of 0.3 (≈30%) is simulated which is manifold higher than the SAR estimates of ≈5%. Thus,

please clarify with example the calculations.

Response: Because we have no observational evidence about BC deposition and associated albedo reduction on real glacier surface, we calculate "sensitivity of glacier runoff to albedo change" using energy-mass balance model. And then we multiply it by albedo reduction estimated from simulated BC deposition. In the original study of the calculation configuration (Fujita, 2007), to avoid influences of changes in meteorological variables, it was assumed that the glacier surface was dusted (and then darkened) by reducing albedo to 0.5 on a given day in the calculation. This study found that impact of the dust fall depended on "when the dust fall event happened". Even if the same albedo reduction happened, the darkened surface during winter will be covered by succeeding snowfall before the melting season while the dust fall event in the mid-monsoon season would result in a significant melting of glacier. Therefore, this study demonstrated a sensitivity test "by changing date of dust fall", and compared consequences of surface albedo and glacier runoff. Also tested was albedo of dusted surface (0.5 to 0.6 or 0.7). The study showed that, even if the dusted albedo was changed, summer mean albedo and annual runoff depth are on an almost linear relation (Fig. 6 of Fujita (2007); shown below).

[Figure]

Figure 6. Summer mean albedo versus runoff depth. Black and grey dots result from dusted surface with an albedo of 0·6 and 0·7 respectively. The result for albedo of 0·5 is not shown

Based on this idea, we have conducted the calculation of this study. However, to reduce calculation load, "dusted date" was changed at 5-days interval. In addition, if the dusted date was end of melting season, it would not affect annual runoff due to a short remaining period. So, we changed the dusted date from the early October to the mid April of a given

year (39 days in total).

In the original study, a single year calculation was made because of data limitation. However, long-term and reliable meteorological data is available so that we perform the calculation for 35 years to reveal impact of glacier surface darkening under different meteorological conditions. Figures showing "summer mean albedo" and "annual runoff depth" averaged for 35 years (error bars are of standard deviations for 35 results). Each point corresponds to a different "dusted date".

We have also now revised Section 2.4 including some more details about simulations with GMB model. Please refer to blue lines in Section 2.4 for the changes in the revised manuscript attached with the Response.

c. What is the X-axis in figure 3C? Is it summer mean albedo or reduction in albedo in the mass balance model (confused with Line 23 page 14). Also include in caption clearly.

Response: The X-axis is the summer mean albedo. Please refer to our response to the previous point. The estimated summer mean albedo and the corresponding annual runoff depth are averaged for 35-years (1979–2014) for each of the nine glaciers. These are plotted in Figure 3c with summer mean albedo on X-axis and annual runoff height on Y-axis. The caption of Figure 3 has been revised for clarity. Please see blue lines for changes introduced (Section 2.4) and Caption of Figure 3 in the revised manuscript attached with the response.

Regarding Line 23 page 14: We have modified the sentence for clarity. Please refer to blue lines for changes introduced (Section 3.2.1). The gradient of linear regression (Figure 3(c), Table 5), which represents an amount of annual glacier snowmelt runoff increase per unit decrease in albedo, is an indicative of the sensitivity of a glacier to albedo change. This gradient is estimated as $-6834$ to $-9945$ mm w.e.y$^{-1}$ per unit of albedo decrease for HKH glaciers under study (refer to Table 5). The annual glacier snowmelt runoff increase per unit decrease in albedo for a cold Tibetan glacier has been estimated as about 3670 mm w.e.y$^{-1}$ (Fujita, 2007; Fujita et al., 2007). This indicates the HKH glaciers are about 2–3 times more sensitive to albedo change (i.e. a higher rate of glacier runoff increase per unit decrease in albedo) than the cold Tibetan glacier.

d. Elaborate more on how ARI is calculated using SAR estimates (Line 9 page 10).

Response: Please also refer to our response to point 2a. Please also see blue lines for changes

introduced (Section 2.4) in the revised manuscript attached with the response.

The results of annual glacier runoff depth and summer mean albedo so obtained is then used to generate the data set for values of SAR and the corresponding annual runoff increase (ARI), estimating the respective values with-respect-to the control run for each of the nine glaciers. We have now included the text "with-respect-to the control run" for clarity. The BC-induced SAR values estimated for each of the glaciers under study is then interpolated with the above data set to estimate the corresponding range for ARI, which is analyzed in Section 3.2.1.

3. The spatial resolution of the models used in the study is relatively very coarse compared to the spatial extend of the glaciers which affects the life cycle of BC estimates due to misrepresentation in terrain and surface emissions. This can contribute a lot to the differences seen in the observation and model values as well as favors the better performance of PA approach than RA approach because of inclusion of vertical profiles. Please include appropriate discussions in relevant places (like $\approx$ Page 11 Line 5).

Response: We have now improved the text and included more discussions in the manuscript related to this. Please see blue lines in Section 3.1.2, page 12–13 in the revised manuscript attached with the response.

At HA stations of HKH (relatively remotely located and mostly influenced by transport of aerosols), the prediction of the three $freesimu$ models are in more sanguinity with the field measurements at HA than that at LA. Further, among the three $freesimu$ estimates, especially that of GCM-indemiss (which is also at a higher spatial resolution than rest other models) has the highest conformity with the measurements at HA stations, hence we utilize BC concentration simulated in GCM-indemiss to estimate BC concentration in snow and BC-induced SAR and their impact over the HKH glaciers. Also, the NMB of $freesimu$ from GCM-indemiss for HA is within the estimated one-sigma uncertainty in mean model simulated atmospheric BC burden. It may be noted that HA stations are accompanied with a feature of a relatively lower magnitude (0.08 $\mu$g m$^{-3}$ to 0.4 $\mu$g m$^{-3}$) and spatial variability of measured BC concentration compared to LA stations (1.1 $\mu$g m$^{-3}$ to 6.5 $\mu$g m$^{-3}$). This feature is also seen in $freesimu$ of GCM-indemiss estimates, with the estimated atmospheric BC concentration over the HKH glaciers being as low as 0.251–0.253 $\mu$g m$^{-3}$ over the Bara Shigri (northern Himalaya)/Zemmu (eastern Himalaya) glacier to as large as 0.517–0.532 $\mu$g m$^{-3}$ over the Chorabari/Gangotri (northern Himalaya) glacier. Hence, our analysis of the

model estimates indicate that *freesimu* estimates of GCM-indemiss used to evaluate BC distribution for HA stations and HKH glaciers, represent reasonably consistently the relative degree of transport of BC and thereby the feature of spatial and seasonal BC distribution across the HKH sites, including the HKH glaciers.

Although, to reduce further the NMB of GCM-indemiss estimates with-respect-to the respective observations over the HKH HA stations and to obtain a more accurate magnitude of BC concentration over the HKH glaciers, it is required to examine BC transport simulation in the *freesimu* of the chemical transport model (CTM) with a better resolved fine grid scale transport processes and a vertical distribution. This requirement is indeed justified (as also mentioned by the Reviewer) as the improved prediction of *constrsimu* is estimated for BC concentration from the PA compared to that from the RA, with the consideration of vertical profile of BC aerosols and comparison of model values near approximate height corresponding to the field measurements at LA stations (please refer to Figure 2c, Table 4, and Section 3.1.1). The setting up of a simulation experiment with a better spatially and vertically resolved BC transport processes with the *freesimu* of a CTM for the HKH region is under progress, with an implementation also of the improved and the latest BC emissions over the Indian region as an input into the model. Results from this simulation will be presented in a future study.

Thus, discrepancies between model and measurements at HA stations is likely attributed to uncertainties which stem out from instrument error, analytical errors, detection capability (pertaining to measurements of low values of atmospheric BC concentration), including degree of accuracy of model processes governing the atmospheric residence time of BC aerosols over HA stations (pertaining to *freesimu* simulations). In summary, our analysis shows that model estimates (*freesimu* of GCM-indemiss over HA stations and HKH glaciers and *constrsimu* over LA stations) with the estimated uncertainty (as large as 45%), represent adequately, in general, the feature and magnitude of BC distribution over HKH sites.

4. A major concern is that the study uses coarse model simulations and offline approaches to estimate SAR, but, eventually uses it to compare and contrast the ARI at very fine spatial (glacier) level. In this view, the quantitative differences reported in this study in estimates of ARI, SAR, BC between the glaciers may be very uncertain. The authors have mentioned about the uncertainties in various steps used in this study for example uncertainties in BC in atmosphere estimates (45%), BC in snow estimates (50–70%) and SAR estimates (20–30%). In a sense, all these processes contribute serially to calculation of increase in snowmelt

runoff. Moreover, there is uncertainties involved in the process of calculating snowmelt runoff increase from glacier mass balance model also. At the same time, the variability in ARI of various glaciers is within 150 mm w.e. $y^{-1}$ (Figure 3b), which is $\approx 50\%$ of the mean ARI over Himalayas. Please include a discussion on these issues towards end of the study.

Response: Thank you for the comment. As suggested by the Reviewer, we have now also included a discussion towards the end of the study. Please see blue lines (page 19–20; also Conclusion, page 23) for changes introduced in the manuscript.

Please note that Figure 3b consist of the box-whisker plot for each of the glaciers. It is to be noted that for this plot, the whisker denotes the maximum and minimum values of the mean ARI for a glacier (due to BC-induced SAR) and the range of box consists of 50% of the ARI data. These details on box-whisker plot is now also provided in the caption of Figure 3 in the revised manuscript.

The mean BC-induced ARI across the glaciers is in the range 135–321 mm w.e.$y^{-1}$, with the highest (lowest) being for Pindari (Sonapani). The variability in ARI across them due to BC-induced SAR is estimated as 58 mm w.e.$y^{-1}$. These details are also now included in the revised manuscript (refer to blue lines in pages 17–18 in the revised manuscript for changes included). Please also note that we have also provided a sensitivity analysis in the manuscript (Section 3.2.2, Figures 3d and 3e) for the simulated impact of albedo reduction on annual glacier runoff for (i) a single glacier hypsometry at different ERA-Interim grids (Figure 3d) and (ii) different glacier hypsometries at a single ERA-Interim grid (Figure 3e).

Regarding the views of the Reviewer on coarse model resolution and the differences in ARI simulated across glaciers and uncertainty aspects:

Please also refer to our response to point 3. The limitation about the coarse gridded models is acknowledged in the Introduction, line no. 20 "However, the ability of coarse-gridded models to simulate adequately the snow depth and thereby the BC concentration in snow and atmospheric BC radiative forcing is limited (Menon et al., 2010; Ménégoz et al., 2014)." In the present study, as mentioned before, an integrated approach is applied, merging the relevant information from observations with a relatively consistent atmospheric chemical transport model estimates. This is done to spatially map as adequately as possible the estimates of atmospheric BC concentration and BC concentration in snow including the corresponding SAR over the HKH region.

Among the three *freesimu* GCM estimates, the prediction of the GCM-indemiss is found in more conformity with the measured than that of the rest other models (please refer to Table 4, Sections 3.1.1, 3.1.2). The *freesimu* estimates of the GCM-indemiss are, therefore, used in *constrsimu* approach (please refer to Section 2.2). The estimates from *constrsimu* mirrored well (Figure 2c, Table 4) the measurements when implemented for LA stations (which are in close proximity to emission sources).

At HA stations of HKH (relatively remotely located and mostly influenced by transport of aerosols), the prediction of the three *freesimu* models, exhibit a better performance with the field measurements than that at LA. Please refer to our response to the point 3 regarding model performance at HA HKH stations and glaciers.

We believe the heterogeneity in ARI estimated across the glaciers to be reasonably adequate. This is because of the application of the consistent estimates of atmospheric BC from the *freesimu* of GCM-indemiss, which showed the best conformity with observations, and merging these estimates with the relevant information from the available observation over the HKH sites to assess the impact of BC aerosols over the HKH glaciers. Further, the estimated BCc from our study (more details now also included in the manuscript, please refer to Section 3.2 with blue lines in the revised manuscript) 
[revised manuscript text omitted]